# RNAi epimutations conferring antifungal drug resistance are inheritable

Carlos Pérez-Arques ⊕[1,5] ✉, María Isabel Navarro-Mendoza ⊕[1,2,3,5] ✉, Ziyan Xu ⊕[1], Grit Walther ⊕[4] & Joseph Heitman ⊕[1] ✉

Epimutations modify gene expression and lead to phenotypic variation while the encoding DNA sequence remains unchanged. Epimutations mediated by RNA interference (RNAi) and/or chromatin modifications can confer antifungal drug resistance and may impact virulence traits in fungi. However, whether these epigenetic modifications can be transmitted across generations following sexual reproduction was unclear. This study demonstrates that RNAi epimutations conferring antifungal drug resistance are transgenerationally inherited in the human fungal pathogen *Mucor circinelloides*. Our research reveals that RNAi-based antifungal resistance follows a DNA sequence-independent, non-Mendelian inheritance pattern. Small RNAs (sRNAs) are the exclusive determinants of inheritance, transmitting drug resistance independently of other known repressive epigenetic modifications. Unique sRNA signature patterns can be traced through inheritance from parent to progeny, further supporting RNA as an alternative molecule for transmitting information across generations. Understanding how epimutations occur, propagate, and confer resistance may enable their detection in other eukaryotic pathogens, provide solutions for challenges posed by rising antimicrobial drug resistance, and advance research on phenotypic adaptability and its evolutionary implications.

The widespread emergence of antimicrobial drug resistance (AMR) is threatening global health, undermining the past century's advances that revolutionized modern medicine and transformed the treatment of infectious diseases[1]. Understanding the molecular mechanisms that confer and transmit resistance is essential, not only for developing novel antimicrobial therapies but also for diagnosing and countering the rise of resistant pathogens. Traditionally, AMR studies have focused on bacteria and antibiotics, yet the threat extends to eukaryotic microbes as well. Fungal infections are increasing at alarming rates, and therapeutic options are scarce. This has prompted the World Health Organization (WHO) to prioritize research and development into the mechanisms of multi- and pan-resistance in fungal pathogens[2], which has recently been termed fAMR[3].

Antifungal drug resistance typically arises from genetic mutations that compromise the interaction of the drug with its target[4]. These changes in the DNA sequence are stable and follow the laws of Mendelian inheritance. Aneuploidy, chromosomal rearrangements, and copy number variation can also mediate drug resistance, tolerance, heteroresistance, and persistence[5]. However, these genomic changes are unstable and transient, usually reverting once drug selective pressure ceases. But there are other, seemingly silent mechanisms that operate—ones that alter gene expression without changing the DNA sequence. These are called epimutations, and rely upon epigenetic mechanisms such as histone modification, DNA methylation, or RNA interference (RNAi) to cause phenotypic changes[6–9]. Fungal epimutations based on RNAi were first identified in *Mucor* species, showing that small RNAs (sRNAs) silencing the FK506 drug target resulted in

[1]Department of Molecular Genetics and Microbiology, Duke University School of Medicine, Durham, NC, USA. [2]Department of Plant Science and Microbiology, Miguel Hernández University, Elche, Spain. [3]Alicante Institute for Health and Biomedical Research (ISABIAL), Alicante, Spain. [4]National Reference Center for Invasive Fungal Infections, Leibniz Institute for Natural Product Research and Infection Biology, Hans Knöll Institute, Jena, Germany. [5]These authors contributed equally: Carlos Pérez-Arques, María Isabel Navarro-Mendoza. ✉e-mail: carlos.parq@duke.edu; maria.navarrom@umh.es; heitm001@duke.edu

epigenetic, reversible resistance[6]. Similarly, in fission yeast, heterochromatin silencing mediated by histone H3 lysine 9 methylation (H3K9me) can result in caffeine resistance[8]. These epimutations cause transient drug resistance that reverts to susceptibility after several mitotic divisions without the drug. However, whether epimutations can be transmitted across generations—a process known as transgenerational inheritance—and the mechanisms driving their heritability, remain subjects of ongoing debate[10–12].

In this study, we show how spontaneous RNAi epimutations conferring antifungal drug resistance are transmitted to the next generation after sexual reproduction in the human fungal pathogen *Mucor circinelloides*. The inheritance pattern is DNA sequence-independent and non-Mendelian, relying exclusively on sRNA molecules as the determinants of inheritance, uncoupled from other repressive epigenetic modifications.

## Results

### RNAi epimutations in opposite mating types of *M. circinelloides*

RNAi epimutations conferring fAMR were first reported in *M. circinelloides* formae *lusitanicus* and *circinelloides*[6], recently identified as separate phylogenetic species (PS10 and PS14)[13]. However, neither species can produce viable progeny through sexual reproduction, presenting a challenge to study genetic and epigenetic inheritance. Therefore, we studied epimutations in two fertile, opposite mating types (− and +) of *M. circinelloides* phylogenetic species 15 (referred to as PS15). PS15 was selected as it is the only species among 16 in the *M. circinelloides* complex able to complete the sexual cycle under laboratory conditions[13]. Despite this advantage, epimutation, RNAi proficiency, and meiotic recombination had not been assessed in this species, and it was unclear if the observed progeny resulted from bona fide sexual reproduction. To address these limitations, whole-genome assemblies and gene annotations for both opposite mating types (PS15− and PS15+) were generated with Oxford Nanopore Technologies (ONT) long reads (Supplementary Table 1). Each genome assembly consists of ~37 Mb distributed among 14 and 15 contigs, showing an N50 of 3.29 and 2.90 Mb for PS15− and PS15+, respectively. Both assemblies exhibit a high level of completeness, with 97.3% of conserved eukaryotic genes present. Protein-protein similarity searches identified homologs of the principal RNAi components involved in establishing and maintaining epimutations[6,14–16] (Supplementary Fig. 1a). sRNA sequencing of both PS15− and + confirmed the presence of canonical small interfering RNAs (siRNAs), characterized by discrete lengths of 21–24 nt and predominantly a 5′ uracil[16] (Supplementary Fig. 1b).

After confirming RNAi proficiency, RNAi-targeted *fkbA* epimutants were isolated by challenging PS15− and + fungal spores with FK506, both alone and in combination with rapamycin (Fig. 1a and Supplementary Fig. 1c, d). Both drugs target FKBP12 encoded by the *fkbA* gene[17], causing similar fungistatic effects: FK506 enforces yeast-like, decreased growth by inhibiting the calcineurin pathway, whereas rapamycin compromises growth by inhibiting TOR signaling (Supplementary Fig. 1c). This dual drug approach enabled efficient screening for loss of FKBP12 function. By this approach, 14 FK506-resistant isolates were obtained after five days of continuous exposure to the drug (Supplementary Fig. 1d). Seven PS15− (E1− to E6− and M1−) and six PS15+ isolates (E10+, E12+, E15+, M1+ to M3+) were selected for their resistance to FK506 and rapamycin (Fig. 1b, Supplementary Fig. 1e, and Supplementary Table 2), indicating loss of FKBP12 function. Sanger-sequencing revealed *fkbA* missense mutations in four isolates, M1− and M1+ to M3+ (and so named Mutants, Supplementary Fig. 1f). No DNA sequence changes were detected within the *fkbA* coding or untranslated regions of the remaining nine isolates, E1− to E6−, and E10+, E12+, and E15+ (Supplementary Fig. 1f). Following mitotic passage without drug, these resistant isolates reverted to wildtype susceptibility, indicating resistance is unstable and transient (Fig. 1c). Because a similar

unstable resistance due to RNAi was previously reported in other *Mucor* species[6,18], northern blot analyses were conducted, revealing antisense sRNAs targeting *fkbA* in these unstable resistant colonies during active epimutation, i.e. before their reversion. These sRNAs were undetectable in wildtype and *fkbA* mutant negative control strains, suggesting epimutation-driven resistance (Supplementary Fig. 1g, Epimutants E1− to E6−, E10+, E12+, and E15+).

RNAi epimutations were further analyzed by sRNA and ribosomal RNA (rRNA)-depleted RNA sequencing. sRNA-producing loci were identified in PS15 naïve wildtype strains, revealing that sRNAs derive primarily from genes and repeated sequences (Supplementary Fig. 2a and Supplementary Data 1), and the most active siRNA loci arise from clusters of repeated elements (Supplementary Fig. 2b). Low levels of sRNAs were detected across the *fkbA* locus in wildtype strains (Fig. 1d, e, and rescaled view in Supplementary Fig. 2c). However, these sRNAs lack canonical siRNA features, such as enrichment in lengths from 21-24 nt or 5′-uracil bias, and are predominantly sense to the *fkbA* mRNA (Supplementary Fig. 2c, d), compatible with transcript degradation[16]. In contrast, the resistant isolates showed an accumulation of *fkbA* antisense siRNAs accompanied by a marked decrease in *fkbA* mRNA levels (Fig. 1d, e, Epimutants), consistent with posttranscriptional gene silencing. Upon removal of drug selective pressure, revertant isolates lost siRNAs and *fkbA* mRNA levels were restored to wild-type values (Fig. 1d, e, Revertants). Our results demonstrate that RNAi-epimutations conferring FK506 and rapamycin resistance are not limited to *M. lusitanicus* and *M. circinelloides* PS14, but are also reproducibly induced in *M. circinelloides* PS15, highlighting the widespread relevance of this adaptive mechanism across the *M. circinelloides* species complex.

### Non-Mendelian inheritance of epigenetic drug resistance after sexual reproduction

Previous studies[6,18] and this work show that RNAi epimutations in *Mucor* species are stable enough to be transmitted through multiple rounds of mitotic division before reversion. Therefore, we hypothesized that epimutations might be similarly transmitted through meiosis and sexual reproduction without drug selective pressure. To test epimutation heritability and determine the genetic basis of the resistance trait, we obtained $F_1$ progeny from a series of genetic crosses. First, an epimutant and a wild type were crossed, utilizing epimutants from both mating types (Fig. 2a, b). Second, two epimutants from opposite mating types were crossed (Fig. 2c). After successful mating of haploid, opposite-mating-type mycelia, the reproductive structures known as zygospores were formed[19]. Zygospores from each cross were individually dissected and germinated after a dormancy period of 2 to 8 weeks, exhibiting germination rates from 1 to 2%. Within each zygospore, haploid nuclei fuse into a diploid nucleus that undergoes meiosis[20,21]. Only one of the four meiotic products is mitotically amplified to form a germsporangium containing genetically identical germspores[22,23], which were considered $F_1$ progeny.

Drug susceptibility to FK506 alone (Supplementary Fig. 3a-c) and in combination with rapamycin (Fig. 2a–c) was assessed in the $F_1$ progeny without any intermediate passaging. Resistance was considered inherited if observed in both conditions. This implies the resistance was present in the progeny prior to the drug challenges, rather than arising from two independent de novo epimutations due to drug exposure. The cross with the − mating type epimutant yielded 6 out of 22 resistant progeny (Fig. 2a), which is inconsistent with a 1:1 inheritance ratio expected from Mendelian inheritance of a monogenic trait in haploid organisms [goodness-of-fit (GOF) Chi-square test, *p*-val = 0.03301]. Similarly, the + mating type epimutant yielded a 2 out of 11 resistance inheritance ratio (Fig. 2b, GOF Chi-square test, *p*-val = 0.03481). Together, these results indicate that resistance can be inherited from either mating type at similar rates, ruling out mating

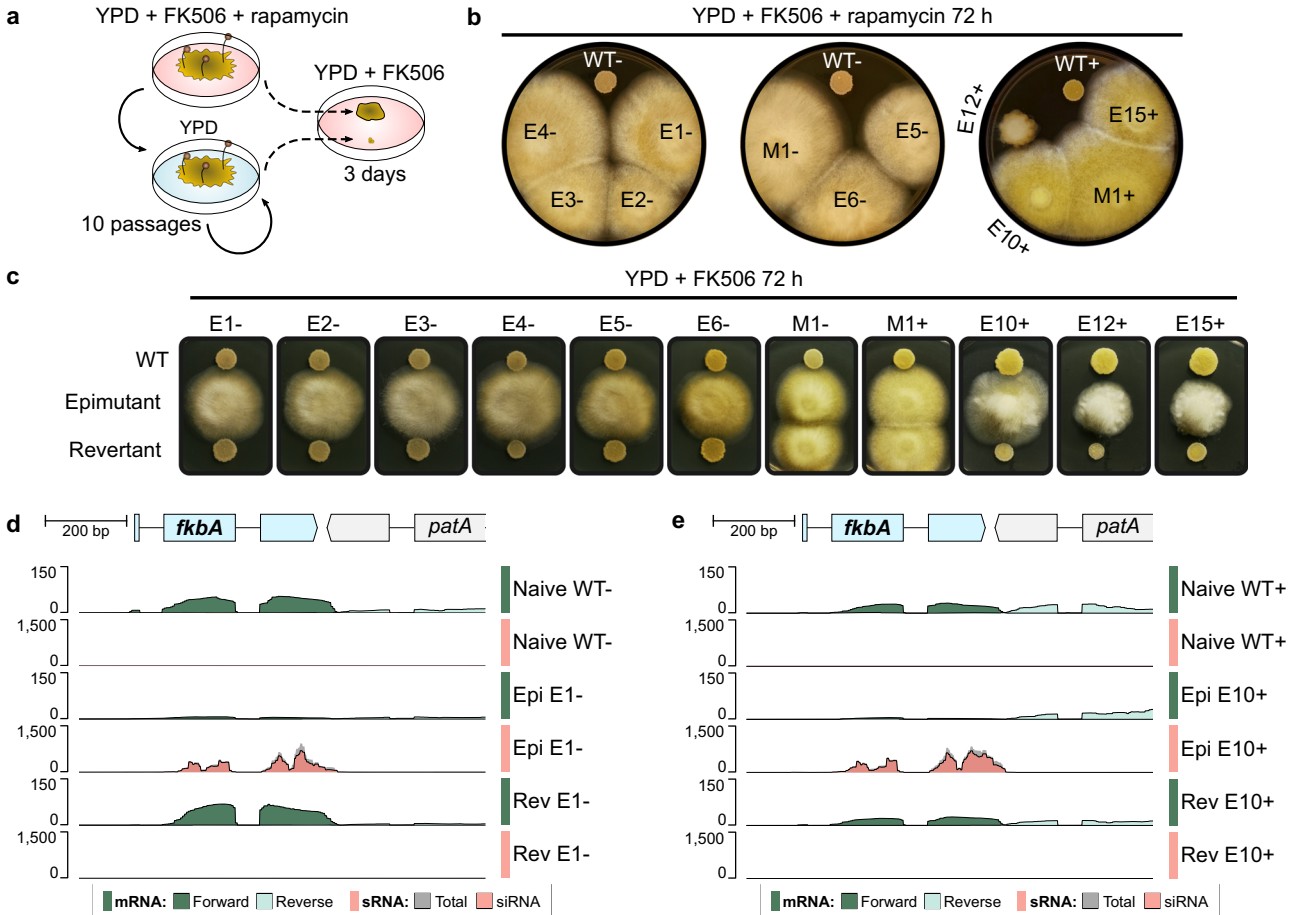

**Fig. 1 | Two opposite mating types from *Mucor circinelloides* Phylogenetic Species 15 (PS15) exhibit RNAi-based epigenetic resistance to FK506 and rapamycin. a** Experimental design to validate FK506 reversible resistance due to loss of FKBP12 function. Briefly, epimutations were induced by exposing wild-type, naïve spores to FK506 for 5 days, as shown in Supplementary Fig. 1d. Resistant mycelium colonies were transferred to FK506 and rapamycin medium to confirm loss of FKBP12 function. Confirmed resistant isolates were serially passaged in drug-free medium for ten cycles of vegetative growth. Spores collected from the final passage were rechallenged with FK506 to assess resistance stability compared to the original resistant isolate. **b** Isolates from PS15 − and + mating types exhibiting FK506 resistance were challenged with both FK506 and rapamycin for 72 h, comparing growth to wildtype strains (WT). Isolates resistant to both drugs are labeled in black, and susceptible isolates in white. Resistant isolates with wild-type *fkbA* DNA sequence are labeled as Epimutants (E); those harboring *fkbA* mutations are labeled as Mutants (M). **c** Epimutants and mutants were passaged on non-selective medium for ten 84 h passages (revertants) and transferred onto FK506-selective medium with wildtype susceptible and resistant control isolates to assess reversion. **d**, **e** Genomic plots of the *fkbA* locus (1 kb) depicting the *fkbA* gene (light blue, gene model PS15m_008921) and 3′ region of the neighboring *patA* gene (light gray). Stranded messenger RNA (mRNA) and small RNA (sRNA) coverage tracks are color-coded for a representative Epimutant, and its corresponding WT control and Revertant isolate from the (**d**) − and (**e**) + mating type. Stranded mRNA coverage was aligned to the forward and reverse strands and overlaid in different shades of green. sRNA coverage shows all aligning reads (total, dark gray) overlaid with reads exhibiting typical small interfering RNA features (siRNA, red), i.e., antisense 21–24 nt reads with a 5′-uracil. Each track represents data from one biological replicate.

type bias or uniparental inheritance. Importantly, resistance was inherited at significantly lower ratios than those expected from Mendelian inheritance, supporting RNAi epimutations as epigenetic traits. The remaining progeny showed wild-type susceptibility to FK506, suggesting the epimutation was either lost or not inherited. On the other hand, crossing two epimutants yielded a higher frequency of drug-resistant progeny (seven out of nine) but still inconsistent with a 100% inheritance expected from a Mendelian monogenic trait present in both haploid parents (Fig. 2c, GOF Chi-square test, p-val = 0.01776). Control crosses between two naïve wildtype isolates yielded only susceptible progeny (eight out of eight, Supplementary Fig. 3d). Similarly, F₁ progeny from an *fkbA* mutant showed resistance ratios consistent with Mendelian inheritance (three out of five, Supplementary Fig. 3e), demonstrating that mutations and epimutations affecting *fkbA* show distinct inheritance rates.

Molecular analysis demonstrated that the progeny isolated from these crosses resulted from bona fide sexual reproduction, including meiosis. We analyzed genetic variation across the PS15− and PS15+

genomes, identifying a total of 18,731 single-nucleotide variants (SNVs) across 13 genomic blocks that exhibited full synteny and contiguity in both opposite mating types (Supplementary Fig. 4). These synteny blocks include six telomere-to-telomere, complete chromosomes and seven contigs, with nine annotated centromeres. By identifying inherited genetic variation across these synteny blocks, we were able to show meiotic recombination events and independent chromosome assortment in all of the F₁ progeny isolates (Fig. 2a–c and Supplementary Fig. 5).

The distribution of the inherited SNVs revealed the source of the *fkbA* allele in the progeny. Resistant progeny inherited the *fkbA* locus at ratios consistent with Mendel's laws (5:3, Chi-square test, p-val 0.4795). The results demonstrate that the epimutation resistance is not genetically linked to the *fkbA* locus, as half of the progeny that inherited resistance from the epimutant parent inherited the *fkbA* allele from the naïve wildtype parent (Fig. 2d, asterisks). Similarly, susceptible progeny inherited the *fkbA* allele from either parent, regardless of their susceptibility. Whole-genome analysis of parents and progeny

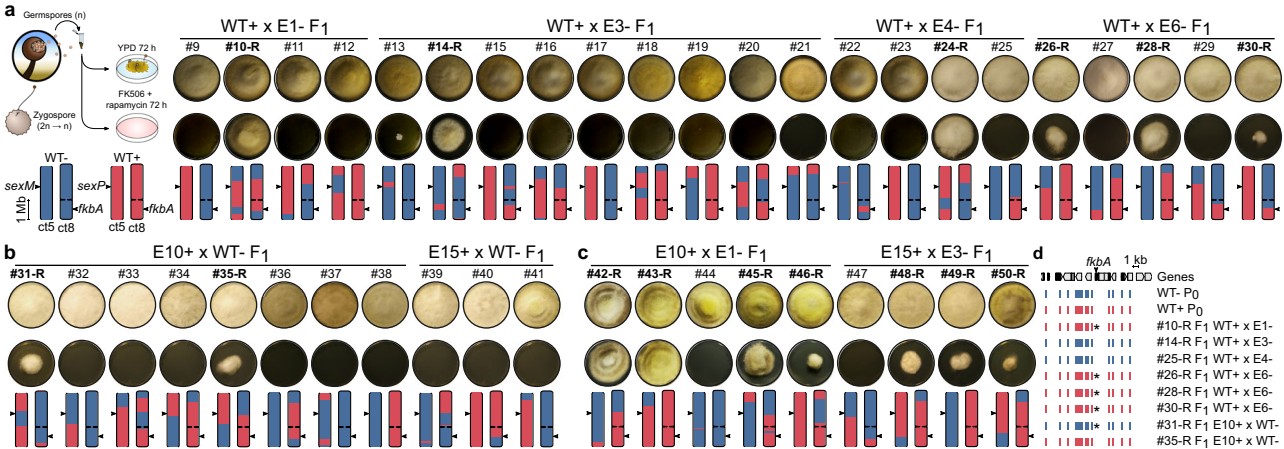

**Fig. 2 | FK506 resistance displays non-Mendelian inheritance ratios in progeny from sexual reproduction.** *M. circinelloides* PS15 progeny from sexual crosses are shown: **a** − mating type epimutant crossed with + mating type wildtype strain (*n* = 22), **b** + mating type epimutant crossed with − mating type wildtype strain (*n* = 11), and **c** + mating type epimutant crossed with − mating type epimutant (*n* = 9). Growth on YPD as a viability control and on FK506+Rapamycin medium is shown for each progeny, along with two contigs containing the *sex* (ct5) and *fkbA* (ct8) loci, indicated by arrows. The contig drawings depict the inheritance pattern of SNVs (− mating type in blue and + mating type in red), demonstrating meiotic recombination and/or independent chromosome assortment. Ct5 is depicted as an incomplete chromosome arm lacking telomeric repeats (open ends), whereas ct8 is a full chromosome depicting a centromere (constriction) and telomeric repeats on both ends (closed ends). **d** Genomic plot showing a 20 kb region centered on the *fkbA* gene. Genetic variation (SNVs) inherited from either mating-type wildtype parent (WT $P_0$) is shown for each resistant (-R) progeny isolated from the crosses of epimutants with wild type.

revealed no changes in the *fkbA* DNA sequence or chromosomal rearrangements contributing to the observed resistance (Supplementary Fig. 6), supporting epigenetic modifications as the exclusive cause of epimutational resistance.

Similarly, in-depth variant calling was conducted to identify off-target mutations that might contribute to FK506 and rapamycin resistance in the epimutant parental strains (E1- to E6−, E10+, E12+, and E15+). Variants were manually curated to identify false positives, discarding variants present in the wildtype negative controls (Supplementary Fig. 7 and Supplementary Data 2). As expected, mutations in *fkbA* were identified in the M1- (TC>T) and M1+ (AGG>A) positive control strains, confirming previous Sanger sequencing results and validating the pipeline. Notably, the epimutant E12+ harbored a substitution (T>C) in a non-coding region upstream of PS15p_212321, which encodes a predicted DNA transposon and is therefore unlikely to be relevant to FK506 and rapamycin epimutational resistance.

**Epimutation inheritance depends on sRNAs, uncoupled from heterochromatin formation**

Substantial evidence supports epigenetic inheritance across a range of eukaryotes, including animals, plants, fungi[12], and other eukaryotic microorganisms[24]. This inheritance mainly relies on chromatin modifications, such as H3K9me, H3K27me, and/or DNA 5-methylcytosine (5mC). Although *Mucor* spp. lack the necessary components for H3K27me and 5mC[25–27], heterochromatin can form and be maintained through H3K9me[26]. Therefore, we hypothesized that RNAi and H3K9me could be coupled to stabilize epimutations, allowing their inheritance in the progeny. This hypothesis was tested by chromatin immunoprecipitation (ChIP) targeting H3K9me2 in PS15 epimutant parents and progeny. H3K9me2-based heterochromatin is abundant in repeats, particularly at centromeres and telomeres (Fig. 3a), consistent with previous findings in *M. lusitanicus* PS10[26]. No H3K9me2 was detected at either the *fkbA* locus in epimutant parents of either mating type or their drug-resistant progeny (Fig. 3a, b and Supplementary Fig. 8a). Instead, abundant siRNAs target *fkbA* in these epimutant isolates, and this results in a decrease in *fkbA* mRNA levels (Fig. 3b and Supplementary Fig. 8a). Despite this reduction, *fkbA* transcription remains active during epimutation as mRNA is still detectable (Fig. 3b and rescaled view in Supplementary Fig. 8b), and RNA polymerase II

(RNAP) binding across this locus does not differ significantly between naïve wildtype and epimutant strains (Fig. 3b–d, Supplementary Fig. 8a, and Supplementary Data 3), indicating that *fkbA* is not transcriptionally silenced during epimutation.

Genome-wide H3K9me remodeling was explored to determine if heterochromatin changes beyond the *fkbA* locus might contribute to FK506 and rapamycin resistance observed in the epimutants. Fourteen genes were embedded in newly gained H3K9me regions in a parent-progeny epimutant pair compared to the wildtype control (Supplementary Fig. 8c, gained H3K9me2, and Supplementary Data 4), but these were not associated with any significant transcriptional changes (Supplementary Fig. 8d and Supplementary Data 5). Similarly, genes that lost H3K9me in epimutants did not show significant expression changes (Supplementary Fig. 8e and Supplementary Data 5). Collectively, these findings indicate that the observed drug resistance is not driven by heterochromatin remodeling in this fungus. Instead, *Mucor* epimutations act posttranscriptionally through an RNAi-dependent mechanism.

These findings were generalized to other *Mucor* phylogenetic species that exhibit RNAi epimutations targeting *fkbA*, *M. lusitanicus* PS10, and *M. circinelloides* PS14[6]. Similarly, we found siRNAs indicative of active *fkbA* posttranscriptional silencing, but no H3K9me2 modification at this locus and no substantial changes in RNAP occupancy between wildtype and epimutant strains (Fig. 3c, d, PS10 and PS14, respectively). In addition to *fkbA*, previously identified epimutations conferring 5-Fluoroorotic acid (5-FOA) resistance by silencing the *pyrG* and *pyrF* pyrimidine biosynthetic genes in *M. lusitanicus* PS10[18] were also tested by ChIP followed by quantitative PCR. Neither *pyrG* nor *pyrF* exhibited any enrichment in H3K9me2 during epimutation (Fig. 3e).

The reduction in *fkbA* transcription during epimutation, consistent with posttranscriptional gene silencing and independent of H3K9me and other chromatin modifications absent in this fungus (H3K27me and 5mC), suggests that siRNAs are functioning as exclusive epigenetic agents of inheritance. We therefore investigated whether siRNAs might be traceable from parents to their progeny. To test this, we analyzed the siRNA profiles and features in a representative set of epimutant parents and their progeny (Fig. 3f). All drug-resistant epimutant progeny harbored abundant sRNAs targeting *fkbA*, which displayed features consistent with canonical siRNAs. These features

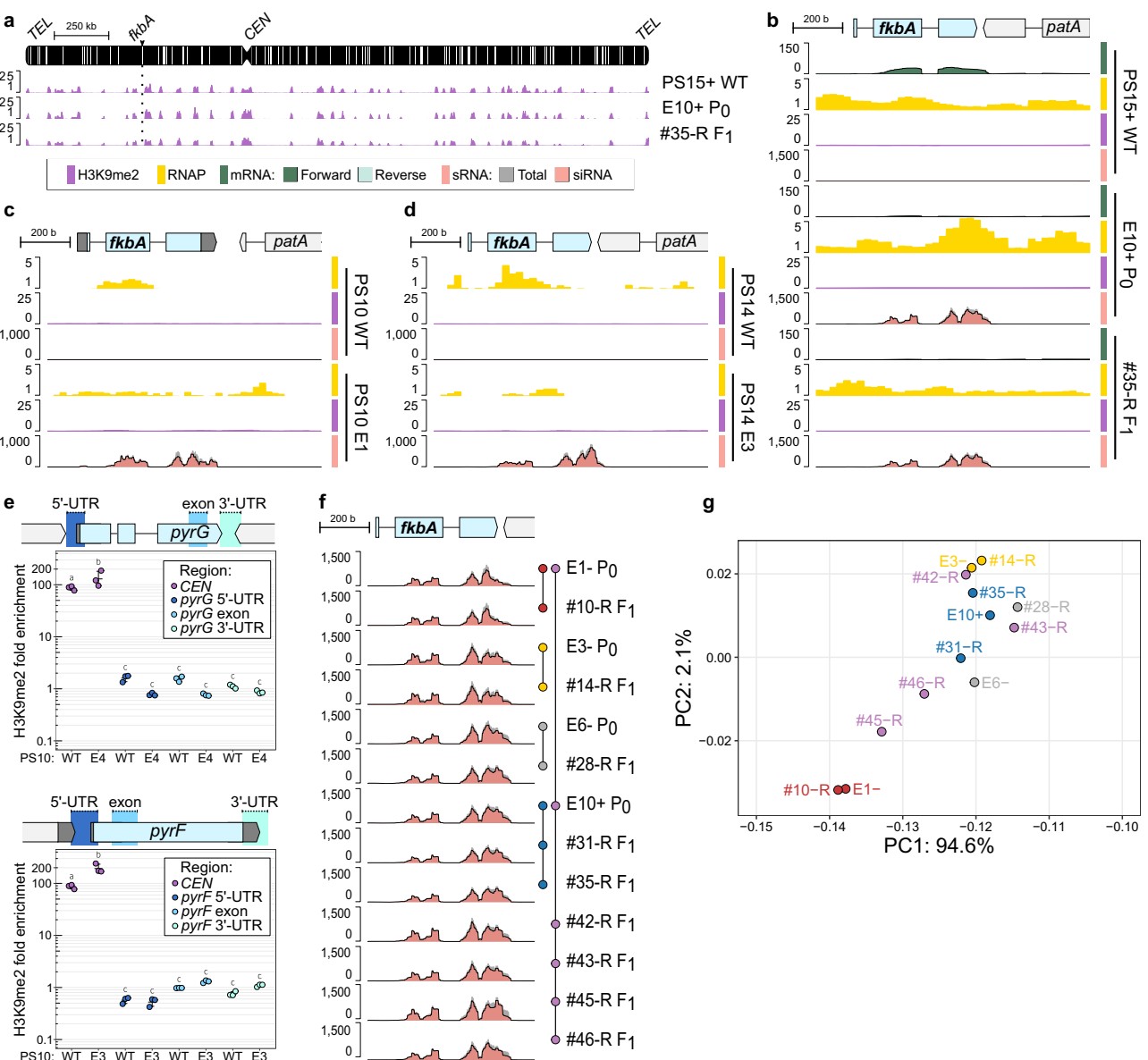

**Fig. 3 | Inherited epigenetic resistance is exclusively driven by RNAi and independent of heterochromatin marks. a–d** Genomic plots showing H3K9me2 enrichment (purple), RNA Polymerase II enrichment (yellow), stranded mRNA (overlaid shades of green for forward and reverse strand), and/or sRNA (dark gray for total sRNA overlaid with red for siRNA) in (**a**, **b**) *M. circinelloides* PS15 wild type, an epimutant parent (P₀) and its F₁ progeny across: (**a**) the full chromosome containing the *fkbA* gene (arrowhead and dotted line). Repeats are depicted as black vertical lines, the centromere as a constriction (*CEN*), and telomeric sequences as rounded ends (*TEL*); (**b**) a magnified view of the *fkbA* locus (1 kb); a wild type (WT) and *fkbA* epimutant (E) from (**c**) *M. lusitanicus* PS10 and (**d**) *M. circinelloides* PS14. Each track represents data from an individual biological replicate. **e** Lack of H3K9me2 ChIP enrichment across *pyrG* and *pyrF* regions is shown for 5-FOA susceptible (WT) and epimutant (E) *M. lusitanicus* PS10 isolates. For each gene, amplicons corresponding to the 5′- and 3′-UTRs and exonic regions were tested by qPCR, as indicated in the gene model. A highly H3K9me2-enriched centromeric amplicon (*CEN*) was also included for comparison. Average values and standard deviations were derived from technical triplicates (*n* = 3). Significant differences were determined using a one-way ANOVA and Tukey HSD tests. Average values not showing significant differences are indicated by the same letter. Source data are provided as a Source Data file. **f** The *fkbA* genomic plot shows total and siRNA coverage for a representative set of epimutant parental (P₀) and F₁ resistant progeny (#-R) isolates. Parents and progeny are represented by colored circles connected by lines. Each track represents data from an individual biological replicate. **g** Principal component analysis of the relative abundance of sRNA sequences complementary to the *fkbA* locus in epimutant parents and progeny shown in (**f**). Progeny from a single epimutant donor cross share the same color as their parent, while progeny from a double epimutant donor cross display mixed colors from both parents (e.g., purple for red and blue parents). Source data are provided as a Source Data file.

include being mainly antisense to the transcript exons, having a defined length of 21–24 nt, and possessing uracil at the 5′ position (Fig. 3f and Supplementary Fig. 9a-c). There is also a notable bias in accumulating siRNAs towards the last exon of the *fkbA* gene.

But more importantly, a striking similarity was observed in the accumulation pattern and relative abundance of siRNAs between epimutant parents and their epimutant progeny (Fig. 3f and Supplementary Fig. 9c). The progeny from two epimutant parents displayed a blend of both parental siRNA profiles, usually favoring one parent. To validate these similarities, Principal Component Analysis (PCA) was conducted on the relative abundances of siRNA molecules complementary to the *fkbA* transcript from each epimutant. Progeny from a single epimutant donor closely clustered with their epimutant parent (Fig. 3g, red, blue, yellow, and gray), whereas progeny from two

epimutants were positioned between the parents, or aligned more closely to one parent (Fig. 3g, progeny in purple, red and blue parents). Taken together, these findings suggest that individual epimutants exhibit a detectable pattern of siRNA, similar to a fingerprint.

The stability of resistance in the progeny was assessed by comparing reversion dynamics and quantifying sRNA profiles in epimutant parents and progeny during subsequent mitotic passaging. As with their epimutant parents, FK506 and rapamycin resistance was maintained under drug selective pressure (Supplementary Fig. 10a). However, the amount of sRNAs did not correlate with the level of resistance, as measured by growth area in drug-containing medium after five passages (Supplementary Fig. 10b). In the absence of selective pressure, resistant isolates typically reverted to drug susceptibility before the tenth passage, following a sharp decline in siRNA levels after the first passage (Supplementary Fig. 10c, d). Therefore, resistance stability, as indicated by the number of passages required for reversion, does not correlate with either the extent of resistance (Supplementary Fig. 10e) or the sRNA levels in resistant isolates (Supplementary Fig. 10f).

## Discussion

DNA is classically regarded as the molecular basis of inheritance, but substantial evidence supports the existence of an RNA world that preceded the current central dogma of molecular biology[28]. In addition to catalyzing chemical reactions, RNA might have stored genetic information and transmitted it across generations. Presently, RNA viruses transmit genetic information by hijacking the synthesis machinery of their hosts, including capsid-free, vertical RNA transmission[29]. But there are more examples beyond viral mechanisms, as there is also transmission of maternal RNA and proteins to oocytes, influencing early zygotic development[30]; and traumatic stress can alter microRNA expression in sperm[31,32]. Our research demonstrates that spontaneous epigenetic drug resistance can be transgenerationally inherited in living organisms, relying exclusively on RNA as the information molecule.

Our current understanding of epigenetic inheritance is mostly based on chromatin modifications, which affect DNA directly, e.g., 5mC, or DNA-binding histone proteins. However, RNA-based epigenetic inheritance remains poorly understood. Nuclear-independent, engineered RNAi epimutation inheritance has been observed across generations in the nematode *Caenorhabditis elegans*[33], but it is challenging to untangle the effects of chromatin modifications because sRNAs guide heterochromatin formation in this organism[34], and many others[35].

Many fungi rely on repressive chromatin modifications—mainly H3K9me, H3K27me, and 5mC—to silence transposable elements and influence development, growth, and virulence in animal and plant pathogens, though their contribution to gene regulation is often difficult to isolate from other genetic factors[9]. Thus far, epimutations as the sole cause of gene expression changes have been convincingly demonstrated in *Mucor* spp. and *Schizosaccharomyces pombe*, where epigenetically driven silencing directly alters a discrete phenotypic trait. In *S. pombe*, caffeine resistance may arise from three independent epimutations based on H3K9me heterochromatin-mediated silencing[8]. Both Clr4-mediated H3K9me and the canonical RNAi pathway are required for this process, consistent with a cotranscriptional gene silencing mechanism. Foundational studies on epigenetic inheritance have shown that heterochromatin at both endogenous and ectopic loci can be transmitted through mitosis and meiosis in *S. pombe*[36–38], especially through inactivation of the histone demethylase Epe1, which mediates a rapid epigenetic resetting mechanism. However, whether these specific epimutations are inheritable following sexual reproduction has not been reported. Interestingly, caffeine induces Epe1 inactivation in *S. pombe*[8], which may complicate its contribution to epimutation stability and heritability. It is also unclear

to what extent either heterochromatin or RNAi contributes, i.e., which is the initial driver of the epimutation.

*M. circinelloides* as a model of epigenetic inheritance may offer a solution to some of these limitations. Recently, we showed that most Mucorales lack the methyltransferases required for 5mC or H3K27me, indicating H3K9me is the primary chromatin modification driving heterochromatin formation[26]. More importantly, RNAi is not required to maintain H3K9me-based heterochromatin at transposable elements, suggesting that these two mechanisms function independently in *M. circinelloides* and *M. lusitanicus*[26]. Our findings build on this and indicate that de novo RNAi epimutations are not triggering heterochromatin formation in either epimutants or their drug-resistant progeny analyzed thus far. These insights enabled us to focus specifically on RNAi inheritance, demonstrating that siRNAs can function as the sole molecular determinants that cause, maintain, and transmit RNAi epimutations, uncoupled from these chromatin modifications commonly associated with epigenetic inheritance in other organisms.

RNAi inheritability is independent of mating type. Notably, it is also independent of the genetic locus targeted by the epimutation, as shown by multiple epimutant drug-resistant progeny that inherited the *fkbA* allele from a naïve, drug-susceptible wildtype parent. Mucoralean sexual reproduction begins with the contact of sexually determined, engorged hyphae called zygophores, which develop into progametangia. As progametangia differentiate into gametangia, they become separated from the mycelium by septae, and fuse to form a zygosporangium and zygospore[19,39]. Plasmogamy and shared cytoplasmic contribution from both gametangia, combined with our findings, suggest cytoplasmic inheritance is the most parsimonious model for siRNA transmission. Cytoplasmic inheritance would explain the higher inheritance rates observed with two epimutant parents, implying that siRNA availability at the tip of the progametangia prior to septa formation is a crucial determinant for inheritance. Previous studies have shown that the RNAi components Dcl2, Ago1, Qip1, and Rdrp2 are essential for initiating and maintaining both epimutations and canonical gene silencing[6,14,15]. Rdrp2 functions as the primary RNA-dependent RNA Polymerase, amplifying secondary siRNAs and thereby propagating RNAi throughout multiple mitotic divisions[40]. However, the role of Rdrp2 in siRNA inheritance is challenging to ascertain as *rdrp2Δ* mutants are unable to mate and produce zygospores[41]. Given that heritable sRNAs in the epimutant progeny harbor typical features of canonical siRNAs, we propose that the canonical RNAi mechanism may extend beyond vegetative growth to preserve epimutations during sexual reproduction, similarly to *C. elegans*[42,43].

Our findings contribute to the growing evidence supporting epigenetic transgenerational inheritance, which allows organisms to adapt rapidly to environmental pressures before genetic changes can occur. Once the environmental pressure is relaxed, epimutations can readily revert, providing remarkable adaptive phenotypic plasticity. This phenomenon is particularly concerning in pathogenic microbes that maintain epigenetic modifications[9,44,45] and/or RNAi machinery[46,47]. In recent years, epimutations have been identified as a novel mechanism that confers fAMR[6,8], posing a significant threat as they may evade detection due to their transient and adaptive nature. Understanding how epimutations are inherited and the mechanisms driving resistance provides solutions to the growing challenges posed by fAMR.

## Methods

### Fungal strains, culture, drug susceptibility testing, and unstable resistant isolation

Supplementary Table 2 lists the fungal strains utilized in this work. All generated strains were derived from *M. circinelloides* CBS394.68 (PS15–) and/or CBS172.27 (PS15+)[13]. To conduct experiments, asexual spores were freshly harvested after 4–6 days of incubation in yeast-peptone-dextrose (YPD) at room temperature. Spores were challenged

with FK506 (1 µg/ml) and/or rapamycin (100 ng/ml) to obtain resistant isolates, test drug susceptibility, and maintain epimutations. Resistant isolates were selected by repeatedly culturing 200 or 2000 spores from naïve, wild-type isolates of both PS15 mating types on FK506-containing medium for 5 days, until sufficient resistant mycelial colonies were recovered. After selection, resistant isolates were challenged with FK506 and rapamycin for 72 h to confirm loss of FKBP12 function in isolates exhibiting resistance to both drugs. To test resistance stability, resistant isolates were serially passaged on non-selective, drug-free YPD for 84 h. Isolates exhibiting complete yeast-like growth on FK506 and therefore, reverting to drug susceptibility after up to ten non-selective passages were classified as unstable resistant isolates.

### Nucleic acids isolation, blotting, immunoprecipitation, and sequencing

Mycelium cultures were collected from YPD–with or without FK506 as needed–, flash frozen in liquid $N_2$ and ground into a powder for nucleic acid isolation. Genomic DNA samples for Illumina sequencing were prepared using Norgen Biotek Yeast/Fungi Genomic DNA Isolation Kit from overnight broth cultures. Genomic DNA libraries were constructed with Roche KAPA HyperPrep Kit. 150-bp paired-end reads were obtained utilizing the NovaSeq X Plus sequencing system. Concurrently, *fkbA* sequences were PCR-amplified using primer pairs listed in Supplementary Table 3 and analyzed by Sanger-sequencing. Ultra-high molecular weight DNA was isolated following a phenol/chloroform-based DNA extraction method[27] from PS15– and PS15+ spores pregerminated for 6 h. DNA was ligated with barcodes (EXP-NBD104, NB10 for PS15– and NB12 for PS15+) and sequencing adapters (SQK-LSK110) to prepare ONT MinION libraries, and loaded into a single MinION flowcell (FLO-MIN106) for a 72 h sequencing run.

Total RNA samples from 72 h cultures in YPD agar were prepared with QIAGEN miRNeasy Mini Kit and divided into small and long RNA preparations. sRNAs were subjected to electrophoresis in urea acrylamide gels (Invitrogen), transferred, and chemically crosslinked [1-methylimidazole 130 mM and 1-ethyl-3-(3-dimethylaminopropyl) 160 mM, pH 8.0, Sigma-Aldrich] to neutral nylon membranes (Hybond NX, Amersham)[48]. *fkbA* sense, α[32P]UTP-labeled radioactive riboprobes were generated using MAXIscript™ T7 Transcription Kit standard procedure (Ambion, Thermo Fisher Scientific Inc.) from PCR-amplified DNA templates (primer pairs listed in Supplementary Table 3). Full-length, radioactive riboprobes were digested into small fragments using an alkaline solution ($NaHCO_3$ 80 mM and $Na_2CO_3$ 120 mM) and hybridized to detect *fkbA* antisense sRNAs, using similarly prepared 5.8S rRNA antisense digested riboprobes as a loading control. Unstable resistant isolates harboring *fkbA* antisense sRNAs were classified as epimutants. sRNA libraries were amplified using QIAseq miRNA library kit, and 75-bp single-end reads were sequenced with Illumina NextSeq 1000 High-Output sequencing system. rRNA-depleted RNA libraries (long RNA) were prepared using Illumina Stranded Total RNA Prep and Ribo-Zero Plus rRNA Depletion Kit using *M. circinelloides* rDNA-specific probes, and 150-bp paired-end reads were obtained with NovaSeq X Plus sequencing system.

Overnight, YPD broth cultures were processed for ChIP following previous research recommendations[26,49], preparing two biological replicates per sample. Briefly, cultures were crosslinked, lysed, and chromatin sonicated throughout 35 cycles 30 s ON/OFF in a Bioruptor (Diogenode) to obtain 100–300 bp chromatin fragments. Sheared chromatin aliquots were stored as input DNA controls. Undiluted ChIP-grade monoclonal antibodies α-H3K9me2 [supplied by Abcam, catalog ab1220, clone mAbcam 1120, lot 1066793-4 of Anti-Histone H3 (di methyl K9) antibody (mAbcam 1220)–ChIP Grade] and α-RNA pol II [supplied by Active Motif, catalog 39497, clone 4H8, lot 23331141 of RNA pol II antibody (mAb)], as well as protein A or G magnetic beads accordingly, were utilized to immunoprecipitate chromatin-bound DNA. After washing and decrosslinking, DNA was purified by a phenol/chloroform-based

extraction method. DNA preparations from the same sample were pooled together. Libraries were prepared using Roche KAPA HyperPrep Kits and sequenced with the Illumina NovaSeq X Plus sequencing system for 150-bp paired-end reads. For quantitative PCR, ChIP and Input DNA were amplified using SYBR Green PCR Master Mix (Applied Biosystems) with specific primer pairs (Supplementary Table 3). qPCRs were performed in triplicate using a QuantStudio™ 3 real-time PCR system. Fold enrichment was determined as a relative quantification to the Input DNA ($2^{-\Delta CT}$) and to the negative control *vma1* values ($2^{-\Delta\Delta CT}$), a constitutively active gene lacking heterochromatin marks.

### Genome assembly, annotation, and homolog identification

Basecalling was performed on MinION raw signal-level, POD5 output using ONT Dorado v0.7.0 (https://github.com/nanoporetech/dorado/) basecaller, selecting the super-accuracy sup@v3.6 basecalling model and the EXP-NBD104 kit (`--kit-name`). Basecalled reads were demultiplexed and trimmed into PS15– and + high-quality FASTQ reads with Dorado modules demux and trim. Draft genome assemblies were generated using Flye v2.9.3-b1797[50] for high-quality reads (`--nano-hq`), setting 0.05 estimated per-read sequencing error rate (`--read-error 0.05`) and excluding alternative contigs (`--no-alt-contigs`). To optimize assembly quality, several runs were performed with varying minimum overlap lengths (`--min-overlap`) ranging from 5000 to 10,000 base pairs in 1000 bp intervals. For each mating type, the assembly exhibiting the highest completeness and contiguity was selected for downstream processing. Genome assemblies were iteratively polished using Racon v1.5.0[51] with default settings until no substantial improvements were observed between successive iterations. Further error correction was performed with default NextPolish1 v1.4.1[52] and 2 v0.2.0[53] using long, ONT-based and Illumina-based, adapter-trimmed short reads. Long reads were aligned with Minimap2 v2.26-r1175[54] default parameters for ONT reads (`-ax map-ont`), as well as short reads with BWA-MEM v0.7.17-r1188[55] default settings, to examine coverage as a measure of assembly correctness. The NCBI contamination detection pipeline was run on the assemblies to identify and remove potential contaminant sequences, including vector, adapter, and non-target organism sequences, following NCBI's standard submission guidelines.

Contig sequences from both mating type assemblies were aligned with Minimap2 (`-x asm5 -N 0 --secondary no`) and visualized using Circos v0.69-8[56] to examine synteny between both assemblies (Supplementary Fig. 4). 13 alignment blocks that showed perfect synteny and contiguity between both assemblies–referred to as synteny blocks–were selected for meiotic recombination analyses.

Repeated sequences were identified utilizing RepeatModeler2 v2.0.5[57] and default parameters, and classified with RepeatMasker v4.1.6 (http://www.repeatmasker.org/) by performing iterative similarity searches against the RepBase database for RepeatMasker (Edition-20181026)[58] and already classified repeats into repeat libraries for each mating type; after that, genome assemblies were masked by RepeatMasker using these repeat libraries. Gene annotation was conducted using the Funannotate pipeline v1.8.15[59] with strain-specific training models generated from our RNA-seq data, including the clean, train, predict, and update modules. Functional annotation of the proteomes was predicted using eggNOG emapper v2.1.12 (`--go-evidence all`) and InterProScan 5.73-104.0[60] (options `--goterms`, `--pathways`, `--iprlookup` enabled). The resulting annotations were integrated into the genome assembly using the Funannotate annotate module.

Genome assemblies were evaluated using QUAST v5.2.0[61] for contiguity metrics and the eukaryotic ortholog database from BUSCO v5.7.1[62] for completeness (`--lineage eukaryota_odb10`, `--mode genome`, `--augustus`). Quality metrics are shown in Supplementary Table 1. RNAi components were identified by NCBI BLASTp v1.12.0[63] searches using *M. lusitanicus* protein sequences as queries, extracted

from M. *lusitanicus* MU402 v1.0 available at the Joint Genome Institute Mycocosm platform (https://mycocosm.jgi.doe.gov/). Identified proteins that also returned positive reciprocal BLASTp hits were retained, and their protein domain configuration was predicted with InterProScan v5.70-102.0 to evaluate their putative function (Supplementary Fig. 1a). Similarly, FKBP12 homologs were identified in both genomes, encoded by the genes PS15m_008921 and PS15p_208735, respectively.

### Mating assays, zygospore isolation, and progeny germination
Opposite mating-type spores were inoculated onto four spots in Whey Agar (WA) plates, allowing them to grow facing each other in dark conditions and at room temperature. After 7 days, zygospores that formed at the intersection of opposite mating types were scraped from the agar using tweezers and suspended in distilled water. Zygospores were germinated according to previous research[13,21,22]. Briefly, the suspensions were successively passed through 40 and 100 μm cell strainers, discarding asexual spores and hyphal fragments and therefore, enriching in zygospores. Single zygospores were manually dissected onto agarose 1% plates at pH 4.0 and incubated at room temperature for 2–8 weeks until germination. Germspores from germinating zygospores were collected in water using a wet micropipette tip.

### Genetic variation analyses
Low-quality and adapter sequences were removed from genomic DNA raw reads using fastp v0.23.4[64] (`--average_qual 20`, `--length_required 148`, `--length_limit 0`, `--trim_poly_g`, and `--dup_calc_accuracy 1`). For subsequent analyses, all samples from either mating type were aligned to the newly generated PS15- genome as the reference unless otherwise stated, to preserve genome coordinate consistency across samples. Processed reads were aligned with BWA-MEM ensuring Picard compatibility (`-M`). Alignments were further processed using the GATK4 suite v4.4.0.0[65] to sort (`gatk SortSam --SORT_ORDER coordinate`), mark duplicates (`gatk MarkDuplicates`) and assign read groups (`gatk AddOrReplaceReadGroups --RGPL Illumina` and other RG accordingly to ID, library, and sample name). A sequence dictionary for the reference genome was generated using `CreateSequenceDictionary`.

For meiotic recombination analyses, variant calling was performed on PS15– and + wildtype datasets by GATK4 `HaplotypeCaller` (`--sample-ploidy 1`). Mixed and symbolic variants were excluded using `SelectVariants` (`--select-type-to-exclude MIXED`, `--select-type-to-exclude SYMBOLIC`). Low quality variants were flagged using `VariantFiltration` with the following arguments: `--filter-expression "QD < 20.0" --filter-name "QD20"`, `--filter-expression "QUAL < 30.0" --filter-name "QUAL30"`, `--filter-expression "SOR > 3.0" --filter-name "SOR3"`, `--filter-expression "FS > 60.0" --filter-name "FS60"`, `--filter-expression "MQ < 40.0" --filter-name "MQ40"`, and filtered out along with variants shared between PS15+ and PS15– using `SelectVariants` (`--discordance $PS15+.vcf`, `--exclude-filtered`). Additionally, variants overlapping repetitive sequences were excluded with Bedtools v2.30.0[66] (`intersect -v`), yielding a high-confidence set of PS15+ specific SNVs from non-repetitive genomic regions. A total of 18,731 SNVs overlapping previously described synteny blocks were identified (`bedtools intersect`), and used to characterize genetic variation in the progeny datasets. To do so, processed alignments from parent and progeny were piled up against this set of PS15+ specific SNVs using GATK4 `GetPileupSummaries`, establishing a threshold of ≥ 95% read support towards either reference (PS15–, REF, `awk '{ratio = ($3 + 0.001) / ($3 + $4 + $5 + 0.001) if (ratio > 0.95) {print $1, $2 - 1, $2, "REF", ratio, "."}}')` or alternate position (PS15+, ALT, `awk '{ratio = ($3 + 0.001)/($3 + $4 + $5 + 0.001) if (ratio <0.05) {print $1, $2 - 1, $2, "ALT", (1 - ratio), "."}}')` to be

considered inherited SNVs. Same parent variants closer than 50 kb were merged by `bedtools merge` (`-d 50000`) to avoid overplotting during rendering (Supplementary Fig. 5).

To identify potential mutations conferring FK506 and rapamycin resistance, a more stringent variant calling analysis was applied to the processed alignments from parental samples (PS15–, E1– to E6–, PS15 +, E10 +, E12 +, and E15 +). Variants were called using GATK4 `HaplotypeCaller` (`--sample-ploidy 1`, `--max-alternate-alleles 3`) and filtered as described above, but with stricter thresholds: `--filter-expression "QD < 5.0" --filter-name "QD5"`, `--filter-expression "DP < 60.0" --filter-name "DP60"`, `--filter-expression "QUAL < 300.0" --filter-name "QUAL300"`, `--filter-expression "SOR > 1.0" --filter-name "SOR1"`, `--filter-expression "FS > 0.1" --filter-name "FS01"`, `--filter-expression "MQ < 60.0" --filter-name "MQ60"`. Variants failing these filters were excluded along with those shared with either wildtype control (PS15– or PS15+) using `SelectVariants` (`--discordance $wildtypes.vcf`, `--exclude-filtered`), and those overlapping repetitive sequences (`bedtools intersect -v`). The resulting high-confidence variants were manually curated by examining the processed alignment from their corresponding wild type; variants already present in the wildtype reads were considered false positives and discarded (Supplementary Fig. 7).

DNA coverage was assessed with Deeptools2 `bamCoverage` v3.5.4[67] (`--binSize 50`, `--normalizeUsing CPM`, `--minMappingQuality 1`) and coverage plots were centered to approximately whole-genome average coverage, which was assumed to represent a haploid genome (Supplementary Fig. 6).

### RNA-sequencing analyses
Raw sRNA reads were processed with UMI-tools (https://github.com/CGATOxford/UMI-tools) to detect the UMI sequences (`umi_tools extract --extract-method=regex, --bc-pattern='.+(?P<discard_1 >AACTGTAGGCACCATCAAT)(?P<umi_1>.{12})')`, and then trimmed with Trim Galore! v0.6.10 (https://github.com/FelixKrueger/TrimGalore) using stringent parameters (`--stringency 4`, `--quality 20`, `--max_n 0`, `-e 0.1`, `--length 10`, `--max_length 75`) to avoid random 3'-end trimming. Processed reads were aligned using Bowtie v1.3.1[68] (`-v 0 -k 1 --best`). ShortStack v3.8.5[69] was run on the resulting alignment files to identify sRNA-producing loci agnostically (`ShortStack --genomefile $PS15-_genome_assembly.fasta --nohp --dicermin 21 --dicermax 25`). Bases of sRNA-producing loci overlapping coding, intergenic, and repetitive regions were quantified using `bedtools intersect -wo` and visualized as a pie chart (Supplementary Fig. 2a). Additionally, sRNAs mapping to annotated gene features were quantified using ShortStack as previously (adding `--locifile $PS15-.gff3`).

Alternatively and to assess splice junction read support (Supplementary Fig. 9c), processed reads were aligned with STAR v2.7.11b[70] (`--outSAMprimaryFlag AllBestScore`, `--alignIntronMin 5`, `--alignIntronMax 200`, `--outSAMmultNmax 1`, `--outFilterMultimapNmax 1`, `--outFilterMismatchNmax 0`, `--sjdbGTFfile $PS15-.gff3`, `--sjdbOverhang 22`, `--sjdbScore 5`, `--alignSJoverhangMin 2`, `--alignSJDBoverhangMin 1`).

Samtools v1.10[71] and Bioawk v20110810 (https://github.com/lh3/bioawk) were used to count and filter reads harboring specific siRNA features, namely antisense orientation, length of 21-24 nt, and 5'-U. For reads mapping to features on the forward strand, reverse reads were considered antisense (`samtools view -L $feature_coordinates.bed -F 4 -f 16 | bioawk -c sam '{if(length($seq) >= 21 & length($seq) <= 24 & substr($seq,length($seq),1) == "A") print}')`; for features on the reverse strand, forward reads were retrieved as antisense (`samtools view -L $feature_coordinates.bed -F 20 | bioawk -c sam '{if(length($seq) >= 21 & length($seq) <= 24 & substr($seq,1,1) == "T") print}')`.

Note that in SAM alignments, all sequences are printed in the forward orientation; therefore, aligned 3′-A reverse reads correspond to 5′-U reads. Coverage files were generated using `bamCoverage` (`--binSize 5`, `--normalizeUsing CPM`, `--minMappingQuality 0`) for both total sRNA (non-filtered alignments) and siRNA (filtered alignments). Relative abundances of siRNA reads were computed *(read number/total fkbA reads)*, and this standardized relative abundances were utilized to perform a PCA to assess similarities among samples (Fig. 3g).

Long RNA raw reads were trimmed with fastp as previously described and aligned with STAR (`--outSAMprimaryFlag All-BestScore`, `--alignIntronMin 5`, `--alignIntronMax 200`, `--outSAMmultNmax 100`, `--outFilterMultimapNmax 100`, `--sjdbGTFfile $PS15-.gff3`, `--sjdbOverhang 150`, `--alignS-JoverhangMin 2`). For differential expression (DE) analyses, read counts mapping to annotated features were computed using feature-Counts (`-p`, `--countReadPairs`, `-t 'exon'`, `-g 'gene_id'`, `-F 'GTF' -s 2`, `--fracOverlap 0`, `fracOverlapFeature 0`). Differential expression was then quantified using DESeq2 v1.44.0[72] by applying the `estimateSizeFactors`, `counts`, and `DESeq` functions to the count matrix, followed by a $\log_2$ transformation of counts per million (CPM). Expression differences were considered significant at an adjusted p-value or False Discovery Rate below 0.05 (FDR ≤ 0.05). Normalized expression values were visualized in a heatmap generated with ComplexHeatmap v2.20.0[73] (Supplementary Fig. 8d, e).

For genomic visualization, long RNA alignments were split by strand using Samtools. Forward reads were obtained by merging second-in-pair forward reads (`samtools view -b -f 128 -F 16`) with first-in-pair reverse reads (`samtools view -b -f 80`). Similarly, reverse reads were obtained by merging second-in-pair reverse reads (`samtools view -b -f 144`) with first-in-pair forward reads (`samtools view -b -f 64 -F 16`). Stranded coverage files were generated using `bamCoverage` (`--binSize 5`, `--normalizeUsing CPM`, `--min-MappingQuality 0`).

### Chromatin immunoprecipitation-sequencing analyses

Raw read samples were processed with fastp to remove adapter and low-quality sequences (`--average_qual 20`, `--length_required 51`, `--length_limit 0`, `--trim_poly_g`, `--dup_calc_accuracy 1`, `--detect_adapter_for_pe`). Processed reads were aligned with BWA-MEM (`-M`). ChIP-enriched regions were identified using MACS2 v2.2.9.1[74] (`callpeak --extsize 200`, `--nomodel`, `--gsize 37441900`) with both narrow (`--call-summits`) and broad (`--broad`) peak settings. Differential binding analysis was performed using DiffBind 3.14.0[75], including read quantification and normalization (`dba.count`, `bUseSummarizeOverlaps=TRUE`, `score=DBA_SCORE_NORMALIZED`), contrast definition (`dba.contrast`, `minMembers=2`, `categories=DBA_FACTOR`) and identification of RNA polymerase II binding differences (`dba.analyze`, `method=DBA_ALL_METHODS`) between epimutant and wildtype samples, specifically at the *fkbA* locus (Supplementary Fig. 8a).

Additionally, genes embedded in H3K9me-based heterochromatin (≥90% of their coding sequence) were identified using `bedtools intersect` and H3K9me2 broad peaks (`-f 0.9 -u`). Shared H3K9me2-embedded genes between epimutant and wildtype samples were visualized as a Venn diagram generated by ggVennDiagram v1.5.2[76] (Supplementary Fig. 8c).

For genomic visualization, coverage of ChIP-enrichment was determined as the IP/Input DNA ratio using Deeptools 2 `bamCompare` (`--binSize 25`, `--normalizeUsing CPM`, `--minMappingQuality 0`, `--operation ratio`).

### Statistical information

Inheritance ratios in the progeny were compared to the expected values of monogenic trait inheritance in haploid organisms with only one parent exhibiting the trait (1:1 ratio, $p = 0.5$, 0.5), or both parents exhibiting the trait (1:0 ratio, $p = 0.95$, 0.05 to avoid expected frequency = 0 and to account for other biological variation, e.g., spontaneous mutations). Statistical significance was assessed by goodness-of-fit Chi-square tests (1 degree of freedom) with a significance level of 0.05. R syntax and test statistics are available as Source Data (sourceDataChisquareTests.txt).

One-way ANOVA testing was conducted to determine significant differences among ChIP qPCR fold enrichment values ($n = 3$), using Tukey's Honestly Significant Distance (HSD) to avoid false positives due to multiple pairwise comparisons. Tests were performed with a 95% CI, and pairwise comparison results were summarized by a compact letter display using the R package multcomp. Individual fold values and test statistics are available as Source Data (sourceDataFig3g.xls).

### Reporting summary

Further information on research design is available in the Nature Portfolio Reporting Summary linked to this article.

### Data availability

Genome assemblies, gene annotation, and raw ONT and Illumina-based raw FASTQ reads are publicly available under NCBI's Sequence Read Archive (SRA) project accessions PRJNA1168935 (PS15–) and PRJNA1168941 (PS15+); the Whole Genome Shotgun projects have been deposited at DDBJ/ENA/GenBank under accessions JBOZQX000000000 (PS15–) and JBNRQL000000000 (PS15+). The versions described in this paper are versions JBOZQX010000000 and JBNRQL010000000, respectively. The remaining raw sequencing data can be accessed under PRJNA1170303, including small RNA, rRNA-depleted RNA, ChIP, and Illumina whole-genome sequencing. In addition, *M. lusitanicus* PS10 wildtype ChIP and small RNA data were retrieved from the publicly available project accession PRJNA903107 and used to compare our findings. Individual SRA run (SSR) accession numbers are listed in Supplementary Data 6. Source data are provided with this paper, including the raw data underlying bar and scatter plots, as well as uncropped and unprocessed scans of Northern blots. These are organized as multiple labeled Tab-separated values (TSV) or PDF files within separate folders in a compressed Source Data ZIP archive. Source data are provided with this paper.

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

## Acknowledgements

We thank our lab manager, Anna Floyd Averette, for her support. We commend Dr. Devjanee Swain Lenz and Duke's Sequencing and Genomic Technologies Core Facility for their assistance, as well as Thomas Milledge and the Duke Computer Cluster team for their computing resources. We thank Prof. Thomas Petes, Prof. Francisco E. Nicolás, Dr. Sheng Sun, and Dr. Jun Huang for critical reading and insightful suggestions on the manuscript. This study was supported by NIH/National Institute of Allergy and Infectious Diseases grants R01-AI170543, R01-AI39115, and R01-AI050113 awarded to J.H. The funders had no role in study design, data collection and interpretation, or the decision to submit the work for publication. J.H. is Co-Director and Fellow of the CIFAR program Fungal Kingdom: Threats & Opportunities.

## Author contributions

C.P.-A., M.I.N.-M., and J.H. conceptualized the project; C.P.-A. and M.I.N.-M. designed the experiments; C.P.-A., M.I.N.-M., and Z.X. performed the experiments; G.W. provided biological resources and critical advice on mating; C.P.-A. and M.I.N.-M. assembled and annotated the genomes; C.P.-A. and M.I.N.-M. analyzed and interpreted the data; C.P.-A. and M.I.N.-M. generated figures; C.P.-A. and M.I.N.-M. wrote the original draft; C.P.-A., M.I.N.-M., and J.H. edited the original draft; C.P.-A., M.I.N.-M., Z.X., J.H., and G.W. reviewed and agreed on the final manuscript.

## Competing interests

The authors declare no competing interests.
