## [Transparent Peer Review file · Nature Communications]

RNAi epimutations conferring antifungal drug resistance are inheritable

Corresponding Author: Dr Joseph Heitman

Version 0:

Reviewer comments:

Reviewer #1

(Remarks to the Author)

The study by Perez-Arques et al. provides the first evidence that epimutations conferring antifungal drug resistance are meiotically inherited in *Mucor circinelloides*, a member of an important group of human fungal pathogens. Epialleles conferring resistance to antifungals had been previously described in *Schizosaccharomyces pombe* and two species within the *Mucor* genus, but it was unknown if these epialleles could be transmitted to sexual progeny and if so, if transmission depends on siRNAs or chromatin modifications. Previous studies focused on two *Mucor* species that lack a defined sexual cycle. Here, two sexually compatible isolates of *Mucor circinelloides* PS15 were studied. The authors generated whole genome assemblies, confirmed the presence of RNAi machinery, and isolated epialleles conferring resistance to FK506 and rapamycin. Epialleles were transmitted to sexual progeny in non-mendelian ratios. Remarkably, the drug-resistance phenotype could be inherited in trans, suggesting that sRNAs present in the cytoplasm transmit the phenotype to progeny. Repressive histone methylation was not detected at silent loci, suggesting that sRNAs mediate inheritance of drug-resistance in a chromatin-independent manner. The manuscript is well written and the experiments are logically designed with appropriate controls. This is an important paper. The study addresses key questions in the fields of epigenetics antifungal drug resistance (i.e. providing new insights into mechanisms of antifungal resistance). This study will have broad appeal to researchers working in diverse fields and with diverse experimental systems. Together, this work provides rigorous support for the conclusions drawn, with one minor exception detailed below.

Based on the data shown in figure 3, it is not clear if RNA pol II occupancy changes in the epimutants or not. How was pol II enrichment quantified?

Reviewer #2

(Remarks to the Author)

This study by Pérez-Arques et al. makes a significant contribution to understanding RNAi-mediated epimutations and their role in antifungal drug resistance. The authors provide compelling evidence that small RNAs can be transmitted across generations in *Mucor circinelloides*, leading to a pattern of non-Mendelian inheritance that occurs independently of DNA sequence changes. Given the increasing concern over antifungal resistance, this research is both timely and highly relevant. The study is well-executed, employing a combination of genomic, transcriptomic, and epigenetic approaches to characterise RNAi-driven inheritance. A major strength of the work is the use of sexual crosses in *Mucor* to demonstrate that RNAi-based drug resistance is maintained across multiple generations. Importantly, the authors establish that this inheritance is mediated exclusively by sRNAs, with no involvement of chromatin modifications such as H3K9me2. This finding provides a crucial precedent for RNA-based epigenetic inheritance in fungi.

Points for consideration

1. The study convincingly shows that sRNAs are inherited; however, the precise mechanism by which they persist through meiosis remains unclear. Direct evidence demonstrating that sRNAs are present in zygospores before germination would strengthen the findings. Sequencing sRNAs from isolated zygospores could provide insight into their stability and potential role during dormancy.

2. The sterility of *rdp2* mutants presents a challenge in fully determining the role of this RNA-dependent RNA polymerase in

epimutation inheritance. Since rdp2 is essential for amplifying secondary sRNAs, its exact contribution to transgenerational inheritance remains uncertain. Would it be possible to test an inducible or conditional rdp2 mutant to assess its role in sRNA transmission during meiosis?

3. Some epimutants (e.g., E10 and E12) exhibit different levels of drug resistance. Is there a correlation between the quantity or position of sRNAs and the extent of resistance?

4. Understanding the baseline sRNA profile in wild-type *Mucor circinelloides* could provide a clearer reference point for the observed epimutations. Could the author provide more info on the sRNA profiling?

5. Following sexual crosses, how stable is the RNAi-based resistance over subsequent generations? Does the resistance eventually diminish, or is it maintained long-term?

Reviewer #3

(Remarks to the Author)

In this study, Pérez-Arques, Heitman, and colleagues investigate the fate of RNAi-based epimutations during meiosis in the pathogenic fungus *Mucor circinelloides*. The results are solid and overall very interesting, but it seems prudent to consider an alternative interpretation, below.

Major concern #1:

1. As the authors have demonstrated, the epimutations are unstable during normal (mitotic) growth, thus, a fraction of epialleles getting "inactivated" during meiosis represents a normal outcome (as meiosis also takes time, etc). The authors were able to measure the rate of such inactivation, being 2/9 (lines 127-130). This parameter can be used to estimate the expected ratios of progeny to test for the Mendelian segregation in a cross between a wildtype and an epimutant parent (E/+).

2. Lines 118-121: for one E/+ cross, the authors obtained 6 "E" (resistant) and 16 "+" (sensitive) progeny. If no correction for the epimutation inactivation is taken (thus expecting 11:11), using R, we get the following:

```
*****  
> chisq.test(rbind(c(6,16), c(11,11)))
```

Pearson's Chi-squared test with Yates' continuity correction

```
data: rbind(c(6, 16), c(11, 11))  
X-squared = 1.5338, df = 1, p-value = 0.2155  
*****
```

This reviewer notes that the p-value obtained above is different from the p-value obtained by the authors under the same assumptions.

Still, if we correct for the fact that a fraction of epimutations is lost, the expected ratio should be $11 - 11 \cdot (2/9) : 11 + 11 \cdot (2/9) = 8.6 : 13.4$. Running the test on these counts gets the following outcome:

```
*****  
> chisq.test(rbind(c(6,16), c(8.6,13.4)))
```

Pearson's Chi-squared test with Yates' continuity correction

```
data: rbind(c(6, 16), c(8.6, 13.4))  
X-squared = 0.26242, df = 1, p-value = 0.6085  
*****
```

The obtained p-value (0.6085) does not allow us to reject the hypothesis that the observed F1 ratio 6:16 deviates from what's expected from a Mendelian segregation of an unstable trait (such as epimutation).

3. Lines 121-122: for second E/+ cross, the authors obtained 2 "E" (resistant) and 9 "+" (sensitive) progeny. Testing for the 1:1 without the correction yields the following:

```
*****  
> chisq.test(rbind(c(2,9), c(5.5,5.5)))
```

Pearson's Chi-squared test with Yates' continuity correction

```
data: rbind(c(2, 9), c(5.5, 5.5))  
X-squared = 1.2644, df = 1, p-value = 0.2608
```

Warning message:
In chisq.test(rbind(c(2, 9), c(5.5, 5.5))) :
Chi-squared approximation may be incorrect

If the same test is done with the correction for the epimutation inactivation (having the expected counts as 4.3:6.7), we obtain the following:

```
*****  
> chisq.test(rbind(c(2,9), c(4.3,6.7)))
```

Pearson's Chi-squared test with Yates' continuity correction

```
data: rbind(c(2, 9), c(4.3, 6.7))  
X-squared = 0.3759, df = 1, p-value = 0.5398
```

Warning message:
In chisq.test(rbind(c(2, 9), c(4.3, 6.7))) :
Chi-squared approximation may be incorrect

Again, the p-value does not allow us to reject the null hypothesis that the epimutant allele actually segregates 1:1 (while also subject to inactivation at the rate of 2/9).

Major concern #2:

Lines 140-148: The authors discover that the epimutation-mediated resistance is not linked to the *fkbA* allele originating from the epimutant parent. Given that the Mendelian segregation of the epimutation-mediated resistance per se cannot be ruled out (above), this result hits at a possibility that epimutation can be controlled genetically by a locus that is not linked to *fkbA*, perhaps being on a separate chromosome. If so, the role of chromatin modifications (and not necessarily H3K9 methylation) remains unclear, since it's not the assayed *fkbA* locus that controls epimutation.

Overall, the data may also be consistent with a model by which a separate Mendelian locus controls the post-transcriptional silencing of *fkbA*. This consideration does not change the main conclusion of the manuscript that "RNAi epimutations conferring antifungal drug resistance are inheritable", but more concerns the mechanism by which the establishment and transgenerational inheritance of epimutations take place.

Reviewer #4

(Remarks to the Author)

The paper entitled "RNAi epimutations conferring antifungal drug resistance are inheritable" tackles the challenge to demonstrate the heritable nature of RNAi epimutations in the context of antifungal resistance, which is a question of major importance and an ambitious objective. The model is the human pathogen *Mucor circinelloides*. The main message conveyed by the study is that spontaneous RNAi epimutations in response to drug challenge can be transmitted to the progeny, inheritance being determined only via smRNAs.

The findings are convincing and the message is quite exciting. The set up may appear simple and straightforward but the work performed is definitely a lot. That being said, there are several points that must be addressed in my opinion.

In a general manner, the materials and methods section is missing many pieces of information to really allow assessing all technical aspects of the paper, and even less to be re-used if one would wish to. For example, parameters used with the various genome assembly and QC tools are mostly missing, the source and version of *M. lusitanicus* proteome is not mentioned, same with the genetic variation analyses and the ChIP-seq analyses. Also, I'm not sure the protocol used to provoke epimutations is crystal clear. The legend of Fig1a must be extended and/or the text clarified. I may not be the only one being a bit uncertain...

Providing details is particularly important in the case of the sRNA sequencing analyses, for which the authors used the Qiagen library system and thus had to apply some specific preprocessing steps to the raw sequencing reads prior use. Similarly, providing the bioawk commands that were used would be welcome for best open science practice (what "specific siRNA features did you use"). This is important since you are using these analyses to make the conclusion that the RNAi machinery works.

Lines 86-87, the authors state that sanger sequencing found genetic mutations in *fkbA* in four isolates out of the 13 tested. However, no information is given regarding what section of the gene was examined, meaning were UTRs checked in addition to cds for example? Non-coding mutations in regulatory elements may still be of influence and the possibility should not be ignored before stating "no DNA sequence changes were found" (lines 87-88).

Regarding the northern blot anti-*fkbA* antisense, I understand from the legend that it was obtained in the Ex isolates BEFORE their reversion, which makes sense and the sentence lines 90-92 should be more explicit. Did you perform the same northern after reversion? Fig 1d and 1e are pretty convincing but not doing the northern when you have all materials to do so does not make sense. And also, there is a typo in the legend where d and e are mislabeled. Nonetheless, I'm mostly ok with your conclusions (maybe the word resulting line 96 is a bit abusive). That being said, I do not understand the last sentence of the paragraph (lines 98 to 100).

In the section dealing with chromatin involvement, the authors state too much from their observations. The title line 149 uses the words “exclusively” and “uncoupled from chromatin modifications”. First, there is more to this world than H3K9. Second, although the demonstration is conclusive on the elements they stated (i.e., H3K9me deposition, genetic mutation in fkbA, and chromosomal rearrangement) the possibility of other contributing mechanisms cannot be excluded, starting with possible genetic mutations elsewhere than fkbA. The “exclusive” wording must be corrected as well as the assimilation of H3K9me to “chromatin modifications”. Same with lines 177-178

The discussion is somehow not really discussing the results. Bringing up more critical views about other chromatin marks or the situation in other fungi (machinery-wise for example) would be a nice addition. The last paragraph starting line 242 is plain useless, as a repetition of info already provided in introduction.

Regarding data availability, the reviewer link failed to work in my hands. But that may be a server glitch. In any case, the “will be” must be replaced by “is” in the text, assuming the author is talking to a reader.

I also have minor comments.

- Extended Figure 1d is too small to see the colonies marked by red arrows;
- No picture of M2+ and M3+ are provided
- Lines 80-81: missing references for the targets of rapamycin and FK506
- Line 115 fig a c is d e
- Line 173 fig 3e is 3d
- Line 125: add p value

Version 1:

Reviewer comments:

Reviewer #1

(Remarks to the Author)

The authors have addressed all of my concerns. This is an exciting study.

Reviewer #2

(Remarks to the Author)

In this revised manuscript, the authors have addressed all of my previous questions satisfactorily. I am happy to recommend it for publication, as the study represents an excellent addition to the field.

Reviewer #3

(Remarks to the Author)

Sure, if the authors believe they must test for 1:1, their analysis makes sense. However, once again, the trait is clearly not purely genetic, since it's not inherited stably even in mitosis (we already know it from the data in the manuscript), so assuming meiotic 1:1 (for a purely genetic trait) is probably not what the authors wish to emphasize anyway.

Reviewer #4

(Remarks to the Author)

I wish to thank the authors for the thorough revision they provided. I found all my comments appropriately addressed, and I gladly recommend the paper for publication in its current form.

REVIEWER COMMENTS

Reviewer #1 (Remarks to the Author):

The study by Perez-Arques et al. provides the first evidence that epimutations conferring antifungal drug resistance are meiotically inherited in *Mucor circinelloides*, a member of an important group of human fungal pathogens. Epialleles conferring resistance to antifungals had been previously described in *Schizosaccharomyces pombe* and two species within the *Mucor* genus, but it was unknown if these epialleles could be transmitted to sexual progeny and if so, if transmission depends on siRNAs or chromatin modifications. Previous studies focused on two *Mucor* species that lack a defined sexual cycle. Here, two sexually compatible isolates of *Mucor circinelloides* PS15 were studied. The authors generated whole genome assemblies, confirmed the presence of RNAi machinery, and isolated epialleles conferring resistance to FK506 and rapamycin. Epialleles were transmitted to sexual progeny in non-mendelian ratios. Remarkably, the drug-resistance phenotype could be inherited in trans, suggesting that sRNAs present in the cytoplasm transmit the phenotype to progeny. Repressive histone methylation was not detected at silent loci, suggesting that sRNAs mediate inheritance of drug-resistance in a chromatin-independent manner. The manuscript is well written and the experiments are logically designed with appropriate controls. This is an important paper. The study addresses key questions in the fields of epigenetics antifungal drug resistance (i.e. providing new insights into mechanisms of antifungal resistance). This study will have broad appeal to researchers working in diverse fields and with diverse experimental systems. Together, this work provides rigorous support for the conclusions drawn, with one minor exception detailed below.

We would like to thank the reviewer for their thoughtful and constructive feedback. The positive comments on the study's design and relevance are greatly appreciated.

Based on the data shown in figure 3, it is not clear if RNA pol II occupancy changes in the epimutants or not. How was pol II enrichment quantified?

We thank the reviewer for highlighting this point. Initially, RNAP occupancy was normalized to input controls, and enrichment was shown as the ratio of IP/Input. We also performed peak calling using MACS2, which identified RNAP peaks at the *fkbA* locus in the epimutant strains, providing evidence that transcription still occurs and supporting our conclusion that epimutation silences

fkbA posttranscriptionally rather than by blocking RNAP recruitment. However, we understand these differences may not have been clearly conveyed.

To address this more rigorously, we have now performed differential binding analysis using DiffBind, which implements both edgeR and DESeq2 for quantification. We conducted new ChIP-seq experiments on isolates of the opposite mating type (wildtype PS15-, epimutant E1-, and its progeny #10; see new Extended Data Fig. 8a), and used the wildtype and epimutant datasets as replicates for differential analysis. Supplementary Data 2 now includes the full analysis across all RNAP peaks, with specific results for the *fkbA* locus highlighted in Extended Data Fig. 8a. This allows for a more robust quantification and p-value assessment of RNAP occupancy at this locus, providing robust evidence supporting posttranscriptional silencing of *fkbA* by RNAi confers epimutational drug resistance.

Reviewer #2 (Remarks to the Author):

This study by Pérez-Arques et al. makes a significant contribution to understanding RNAi-mediated epimutations and their role in antifungal drug resistance. The authors provide compelling evidence that small RNAs can be transmitted across generations in *Mucor circinelloides*, leading to a pattern of non-Mendelian inheritance that occurs independently of DNA sequence changes. Given the increasing concern over antifungal resistance, this research is both timely and highly relevant.

The study is well-executed, employing a combination of genomic, transcriptomic, and epigenetic approaches to characterise RNAi-driven inheritance. A major strength of the work is the use of sexual crosses in *Mucor* to demonstrate that RNAi-based drug resistance is maintained across multiple generations. Importantly, the authors establish that this inheritance is mediated exclusively by sRNAs, with no involvement of chromatin modifications such as H3K9me2. This finding provides a crucial precedent for RNA-based epigenetic inheritance in fungi.

We very much appreciate the reviewer's time and effort in reviewing our manuscript and providing positive and constructive feedback.

Points for consideration

1. The study convincingly shows that sRNAs are inherited; however, the precise mechanism by which they persist through meiosis remains unclear. Direct evidence demonstrating that sRNAs are present in zygosporangia before germination would strengthen the findings. Sequencing sRNAs from isolated zygosporangia could provide insight into their stability and potential role during dormancy.

We appreciate the reviewer's suggestion regarding the demonstration of the presence of sRNA in zygosporangia prior to germination. However, we would like to note the current limitations of our methodology. Despite multiple attempts, we were unable to purify high-quality small RNA from zygosporangia, likely due to the structural resilience of these spores, which hampers current RNA extraction protocols. Furthermore, although our methodology is able to enrich for zygosporangia, it cannot entirely eliminate physical contamination from parental hyphae, complicating bulk RNA analyses directly from zygosporangia.

To strengthen the evidence, we assessed drug susceptibility by two independent challenges using the same batch of germsporangiospores (sexual progeny spores). One, with FK506 alone

(Extended Data Fig. 3a-c), and another with FK506 and rapamycin (Fig. 2a-c). Because resistance emerged in the same isolates under both challenges, this strongly suggests that the resistance phenotype was already present in the progeny prior to drug exposure, rather than arising independently in each challenge. Although these results do not directly demonstrate the presence of sRNAs in zygospores, they provide compelling support that RNAi-driven drug resistance was inherited.

Nevertheless, we appreciate the reviewer's valuable suggestion and plan to direct future efforts toward developing a single-cell small RNA sequencing approach for isolated zygospores. This strategy would minimize contamination and enable a more accurate characterization of sRNA populations across all stages of sexual development in *Mucor*. In this fungus, sexual reproduction is initiated by the fusion of compatible hyphae, leading to the formation of immature and then mature zygospores, which are thick-walled, stress-resistant structures that undergo dormancy. Upon germination, zygospores give rise to a germsporangiophore, a specialized structure that bears a terminal germ sporangium containing progeny spores (germsporangiospores). Profiling sRNAs across these distinct stages (immature and mature zygospores, germinating zygospores, germsporangiophores, and germsporangiospores) would provide key insights into RNA-mediated inheritance during sexual reproduction. Although single-cell sRNA sequencing remains technically challenging, particularly in filamentous fungi where it has not yet been attempted, it represents an exciting direction for future research.

2. The sterility of *rdp2* mutants presents a challenge in fully determining the role of this RNA-dependent RNA polymerase in epimutation inheritance. Since *rdp2* is essential for amplifying secondary sRNAs, its exact contribution to transgenerational inheritance remains uncertain. Would it be possible to test an inducible or conditional *rdp2* mutant to assess its role in sRNA transmission during meiosis?

We truly appreciate the reviewer's suggestion. While understanding the specific roles of different RNAi enzymes is important, this study primarily focuses on investigating whether small RNAs can be inherited and how major chromatin modifications contribute to this process. Unfortunately, PS15 transformation remains a challenge, as auxotrophic strains are not yet available. To explore this further, we would need to generate auxotrophic strains for both mating types, optimize transformation efficiency, generate *rdp2* mutants, cross them with an epimutant, select *rdp2* mutant progeny, and analyze how epimutations are inherited.

This is an exciting direction for future research, and we are working towards these goals. We plan to analyze the involvement of distinct RNAi enzymes, such as alternative Argonautes (Ago2 and Ago3), RdRPs (Rdrp1, Rdrp2, and Rdrp3), Dicer-like enzymes (Dcl1 and Dcl2), and other components (Qip1 and RnhA) in future studies. We also thank the reviewer for suggesting inducible or conditional mutants to bypass the infertility of *rdrp2* deletion mutants. Ideally, we would induce *rdrp2* only during mating and turn it off during meiosis, zygospore germination, and subsequent stages to assess if resistance is inherited or requires Rdrp2. However, we have not yet developed a TET-On-like system for *Mucor* species, which presents another exciting challenge for future work.

3. Some epimutants (e.g., E10 and E12) exhibit different levels of drug resistance. Is there a correlation between the quantity or position of sRNAs and the extent of resistance?

We thank the reviewer for this insightful question. To address it more directly, we have now included a new supplementary figure (Extended Data Fig. 10a, b) showing a quantitative comparison of growth area and sRNA abundance in a range of epimutant isolates, including both parents and progeny. This is now also included as part of the revised text. These data confirm that the extent of drug resistance does not depend on the amount of sRNAs targeting *fkbA*, which remain relatively constant among all of the isolates. This lack of correlation likely stems from the nature of our sampling: we collected mycelium that had already acquired resistance. As such, isolates that may have initially produced sRNAs more slowly (and therefore grew more slowly) would still accumulate similar levels of sRNAs per the same amount of tissue. In other words, a threshold level of sRNAs may be required to support resistant growth, and because we are sampling mycelium that already exhibits resistance, it is reasonable that these samples contain similar amounts of sRNAs.

4. Understanding the baseline sRNA profile in wild-type *Mucor circinelloides* could provide a clearer reference point for the observed epimutations. Could the author provide more info on the sRNA profiling?

We appreciate the reviewer's suggestion and agree that examining sRNA production in a naïve wildtype strain provides valuable insight into the triggers of epimutations.

To address this initially, we analyzed the sRNA population in the wild-type strain and found only minimal levels of sRNAs targeting *fkbA* (Figure 3b). To extend our previous analysis, we have also determined the proportion of the genome producing sRNAs, and whether these derive from gene, repeated, or intergenic sequences (Extended Data Fig. 2a). In line with canonical sRNA

pathways, the detected sRNAs in the wildtype strains predominantly target genomic repeats and genes, suggesting they arise from active transcription. We have also included an assessment of sRNA distribution across an entire chromosome (contig 8 containing the *fkbA* gene) in Extended Data Fig. 2b. To focus on the *fkbA* locus, we provided new panels showing genomic plots with a narrower y-axis scale (Extended Data Fig. 2c) and quantified the presence of canonical siRNA features in sRNA targeting *fkbA* in wildtype, naïve PS15 isolates. No specific enrichment at the *fkbA* locus was detected, indicating that *fkbA* is not a primary target of sRNA production under normal conditions. This suggests that epimutation is selected and amplified upon drug exposure. Whether the residual levels of sRNAs detected, possibly degradation products, might serve as a template for RNAi (and how) will be explored in future research.

5. Following sexual crosses, how stable is the RNAi-based resistance over subsequent generations? Does the resistance eventually diminish, or is it maintained long-term?

We appreciate the reviewer's question. Similar to the progenitor strain, RNAi-based resistance in the progeny is unstable and disappears after a few asexual generations.

To better illustrate this, we have added new supplementary panels (Extended Data Fig. 10c-f) analyzing reversion dynamics as a proxy for resistance stability. We tracked sRNA content across successive passages until complete reversion to susceptibility was achieved in both parents and progeny, obtaining similar results in parental and progeny isolates. The sRNA levels drop sharply after just one asexual generation and remain nearly undetectable. However, this residual sRNA pool was sufficient to reamplify the epimutation upon re-exposure to FK506 in some cases. Complete reversion, defined as loss of growth after three days in FK506, occurs several passages after sRNAs become undetectable. Interestingly, the number of passages required for reversion did not correlate with either the initial level of drug resistance or the initial amount of sRNAs.

Reviewer #3 (Remarks to the Author):

In this study, Pérez-Arques, Heitman, and colleagues investigate the fate of RNAi-based epimutations during meiosis in the pathogenic fungus *Mucor circinelloides*. The results are solid and overall very interesting, but it seems prudent to consider an alternative interpretation, below.

Major concern #1:

1. As the authors have demonstrated, the epimutations are unstable during normal (mitotic) growth, thus, a fraction of epialleles getting “inactivated” during meiosis represents a normal outcome (as meiosis also takes time, etc). The authors were able to measure the rate of such inactivation, being 2/9 (lines 127-130). This parameter can be used to estimate the expected ratios of progeny to test for the Mendelian segregation in a cross between a wildtype and an epimutant parent (E/+).

Thank you for your valuable suggestions regarding the fraction of epimutation inactivation during meiosis. We appreciate your perspective on using this parameter to estimate the expected ratios of progeny for testing Mendelian segregation in a cross between a wildtype and an epimutant parent (E-/ + or E+/-). However, we respectfully disagree that using the 2/9 inactivation rate, which derives from another cross involving two epimutant parents (E-/E+), is the best approach to correct the expected values for our crosses.

The primary goal of our analysis is to determine if a trait is inherited according to Mendelian laws for a monogenic trait in haploid organisms, which predicts a 1:1 segregation ratio in the progeny. Correcting this predetermined and well-established expected value would undermine the purpose of our initial hypothesis and testing. Assuming the predefined 1:1 ratio needs correcting already implies that the trait does not follow monogenic Mendelian inheritance, especially if we need to correct by a 2/9 ratio parameter. Furthermore, we consider that using observed frequencies from one experimental cross (E-/E+) to adjust expected values for all crosses may introduce bias. It is important to establish statistical hypotheses and methods before conducting experiments to ensure methodological soundness. Therefore, using results from one experiment to adjust the statistical analysis of others may not be the most appropriate approach.

Moreover, the loss of epimutation in 2 out of 9 progeny in a cross involving two epimutant parents (E-/E+) may result from various mechanisms, such as epigenetic reprogramming or sRNA loss (i.e., reversion) after growing in non-selective medium during the mating process. Growing in non-

selective medium involves a bottleneck effect that occurs before meiosis takes place, where sRNA content can vary greatly between mycelium sectors. Therefore, the amount of sRNAs in the hyphae upon hyphal fusion that generates a zygosporangium is inherently stochastic and vastly determines if the epiallele is inherited or not. These random fluctuations can significantly influence the progeny epigenetic composition, making it challenging to discern between meiotic and premeiotic mechanisms without further evidence and testing. While quantifying the extent of each of these mechanisms would indeed be valuable, it falls outside the scope of our current study, and it is not necessary to evaluate our initial hypothesis. However, we appreciate the reviewer's insightful suggestions and will consider exploring this in future research endeavors.

Therefore, we respectfully consider that using a 2/9 ratio as a proxy for epiallele inactivation during meiosis would introduce more bias than benefit in our testing. It would also detract from our original hypothesis, which aims to determine if this trait follows a Mendelian inheritance pattern for a monogenic trait.

2. Lines 118-121: for one E/+ cross, the authors obtained 6 “E” (resistant) and 16 “+” (sensitive) progeny. If no correction for the epimutation inactivation is taken (thus expecting 11:11), using R, we get the following:

```
> chisq.test(rbind(c(6,16), c(11,11)))
```

Pearson's Chi-squared test with Yates' continuity correction

data: rbind(c(6, 16), c(11, 11))

X-squared = 1.5338, df = 1, p-value = 0.2155

This reviewer notes that the p-value obtained above is different from the p-value obtained by the authors under the same assumptions.

Still, if we correct for the fact that a fraction of epimutations is lost, the expected ratio should be $11-11*(2/9) : 11+11*(2/9) = 8.6 : 13.4$. Running the test on these counts gets the following outcome:

```
> chisq.test(rbind(c(6,16), c(8.6,13.4)))
```

Pearson's Chi-squared test with Yates' continuity correction

data: rbind(c(6, 16), c(8.6, 13.4))

X-squared = 0.26242, df = 1, p-value = 0.6085

The obtained p-value (0.6085) does not allow us to reject the hypothesis that the observed F1 ratio 6:16 deviates from what's expected from a Mendelian segregation of an unstable trait (such as epimutation).

3. Lines 121-122: for second E/+ cross, the authors obtained 2 “E” (resistant) and 9 “+” (sensitive) progeny. Testing for the 1:1 without the correction yields the following:

> chisq.test(rbind(c(2,9), c(5.5,5.5)))

Pearson's Chi-squared test with Yates' continuity correction

data: rbind(c(2, 9), c(5.5, 5.5))

X-squared = 1.2644, df = 1, p-value = 0.2608

Warning message:

In chisq.test(rbind(c(2, 9), c(5.5, 5.5))) :

Chi-squared approximation may be incorrect

If the same test is done with the correction for the epimutation inactivation (having the expected counts as 4.3:6.7), we obtain the following:

> chisq.test(rbind(c(2,9), c(4.3,6.7)))

Pearson's Chi-squared test with Yates' continuity correction

data: rbind(c(2, 9), c(4.3, 6.7))

X-squared = 0.3759, df = 1, p-value = 0.5398

Warning message:

In `chisq.test(rbind(c(2, 9), c(4.3, 6.7)))` :

Chi-squared approximation may be incorrect

Again, the p-value does not allow us to reject the null hypothesis that the epimutant allele actually segregates 1:1 (while also subject to inactivation at the rate of 2/9).

We appreciate the opportunity to clarify the nature of our Chi-square tests and would like to address reviewer's comments 2 and 3 in a single response as follows. We would like to point out that inputting the observed and expected absolute counts as a dataframe into the `chisq.test` function in R causes it to be analyzed as two independent groups in a 2x2 contingency table. This generates incorrect expected values from row and column totals, leading to incorrect X-squared and interpolated *p*-values. For example, doing this:

```
> chisq.test(rbind(c(6,16), c(11,11)))
```

will result in the data being analyzed as follows:

	Independent Group 1	Independent Group 2	Column totals
FK506 resistant	6	11	17
FK506 susceptible	16	11	27
Row totals	22	22	44

In a 2x2 contingency table chi-square test, the expected values (*E*) are derived from row and column totals using the formula:

$$E_{row,col} = \frac{(\text{Row Total}) \times (\text{Column Total})}{\text{Grand Total}}$$

$$E_{(1,1)} = \frac{(17 \times 22)}{44} = 8.5$$

$$E_{(1,2)} = \frac{(17 \times 22)}{44} = 8.5$$

$$E_{(2,1)} = \frac{(27 \times 22)}{44} = 13.5$$

$$E_{(2,2)} = \frac{(27 \times 22)}{44} = 13.5$$

If we continue with the Chi-square test manually using these values and applying Yates' correction for continuity as the reviewer mentioned:

$$\begin{aligned} \chi^2 &= \sum \frac{(|O - E| - 0.5)^2}{E} \\ \chi^2 &= \frac{(|6 - 8.5| - 0.5)^2}{8.5} + \frac{(|11 - 8.5| - 0.5)^2}{8.5} + \frac{(|16 - 13.5| - 0.5)^2}{13.5} + \frac{(|11 - 13.5| - 0.5)^2}{13.5} \\ &= \frac{(2.5 - 0.5)^2}{8.5} + \frac{(2.5 - 0.5)^2}{8.5} + \frac{(2.5 - 0.5)^2}{13.5} + \frac{(2.5 - 0.5)^2}{13.5} \\ &= \frac{4}{8.5} + \frac{4}{8.5} + \frac{4}{13.5} + \frac{4}{13.5} = 0.47 + 0.47 + 0.30 + 0.30 = \mathbf{1.53} \end{aligned}$$

Calculating degrees of freedom (*df*):

$$df = (\text{rows} - 1) \times (\text{columns} - 1) = (2 - 1) \times (2 - 1) = 1$$

And interpolating the *p-value*:

$$p = P(\chi^2 \geq 1.53, df = 1) \approx 0.216$$

This roughly aligns with the values the reviewer mentioned. We sincerely hope the reviewer agrees that treating observed and expected values as two independent groups in a 2x2 contingency table was not the intention of our analysis and results in artificial and incorrect expected values. Presenting the data this way would be useful for determining if two independent crosses follow the same inheritance pattern, but it is not the most appropriate for our particular study, which aims to determine if the trait segregates according to predetermined Mendelian ratios for a monogenic trait. In this case, a goodness-of-fit (GOF) test is appropriate for the data and is traditionally performed using the Chi-square test to assess if an inherited trait deviates from predetermined Mendelian ratios, as follows:

$$\chi^2 = \sum \frac{(O_i - E_i)^2}{E_i}$$

For example, with our data from the first cross described:

	Observed (O)	Expected (E)
FK506 resistant	6	11

FK506 susceptible	16	11
-------------------	----	----

$$\chi^2 = \frac{(6 - 11)^2}{11} + \frac{(16 - 11)^2}{11} = \frac{(-5)^2}{11} + \frac{(5)^2}{11} = \frac{25}{11} + \frac{25}{11} = \frac{50}{11} \approx 4.545$$

Calculate degrees of freedom:

$$df = 2 - 1 = 1$$

Traditionally, we would examine a critical values of chi-square distribution table (e.g., https://www.mun.ca/biology/scarr/4250_Chi-square_critical_values.html) and check if our χ^2 is significant for a 0.05 probability of exceeding the critical value ($\chi^2 \geq 3.841$), which it does. Or we could interpolate the *p*-value in R:

```
=====
> 1 - pchisq(q = 4.545 , df = 1)
[1] 0.03301502
```

$$p = P(\chi^2 \geq 4.545, df = 1) \approx 0.033$$

We can also do a GOF chi-square test in R by inputting the expected values as a vector of probabilities $p = c(0.5, 0.5)$, instead of absolute values.

```
=====
> chisq.test(x = c(6,16), p = c(0.5,0.5))
      Chi-squared test for given probabilities
data:  c(6, 16)
X-squared = 4.5455, df = 1, p-value = 0.03301
```

=====

As we did for all our crosses:

```
=====
> chisq.test(x = c(2,9), p = c(0.5,0.5))
      Chi-squared test for given probabilities
data:  c(2, 9)
X-squared = 4.4545, df = 1, p-value = 0.03481
```

```

> chisq.test(x = c(7,2), p = c(0.95,0.05))
      Chi-squared test for given probabilities
data:  c(7, 2)
X-squared = 5.6199, df = 1, p-value = 0.01776
Warning message:
In chisq.test(x = c(7, 2), p = c(0.95, 0.05)):
  Chi-squared approximation may be incorrect

```

```
=====
```

It is important to note that we did not apply Yates' correction for continuity, which is specifically designed for 2x2 contingency tables involving small (typically less than 5) expected values and not for GOF tests (Yates, 1934). Additionally, the benefits of using Yates's correction even in 2x2 contingency tables are also in dispute (Conover, 1974; Maxwell, 1976).

We apologize for not doing a better job clarifying the methodology, and we understand how the former lines 427-429 from the Methods section could cause confusion. We have rephrased the Methods section accordingly, addressed the exact Chi-square test we performed in the Results section, and hope the reviewer agrees with our testing.

Conover, W. J. (1974). Some reasons for not using the Yates continuity correction on 2×2 contingency tables. *Journal of the American Statistical Association*, 69(346), 374. <https://doi.org/10.2307/2285661>

Maxwell, E. A. (1976). Analysis of contingency tables and further reasons for not using Yates correction in 2x2 tables. *The Canadian Journal of Statistics / La Revue Canadienne de Statistique*, 4(2), 277. <https://doi.org/10.2307/3315141>

Yates, F. (1934). Contingency Tables Involving Small Numbers and the χ^2 Test. *Journal of the Royal Statistical Society Series B: Statistical Methodology*, 1(2), 217–235. <https://doi.org/10.2307/2983604>

Major concern #2:

Lines 140-148: The authors discover that the epimutation-mediated resistance is not linked to the fkbA allele originating from the epimutant parent. Given that the Mendelian segregation of the epimutation-mediated resistance per se cannot be ruled out (above), this result hits at a possibility that epimutation can be controlled genetically by a locus that is not linked to fkbA, perhaps being on a separate chromosome. If so, the role of

chromatin modifications (and not necessarily H3K9 methylation) remains unclear, since it's not the assayed *fkbA* locus that controls epimutation.

Overall, the data may also be consistent with a model by which a separate Mendelian locus controls the post-transcriptional silencing of *fkbA*. This consideration does not change the main conclusion of the manuscript that “RNAi epimutations conferring antifungal drug resistance are inheritable”, but more concerns the mechanism by which the establishment and transgenerational inheritance of epimutations take place.

We appreciate the reviewer's thoughtful suggestion regarding potential alternative models. As noted, this consideration does not alter the main conclusion of the manuscript.

Because this comment builds on the previous concern regarding Mendelian segregation, we hope the clarification provided there addressed the issue. Nevertheless, the idea that genetic variation at loci unlinked to *fkbA* might influence RNAi-mediated resistance is worth addressing. To explore this possibility, we conducted a variant calling analysis across the genomes of resistant parental strains (Supplementary Data 2) and generated Extended Data Fig. 7, which shows all putative variants after filtering out those present in repeat regions or wildtype controls. Most candidates were deemed false positives based on their presence in wildtype reads (see WT Aln. track) and were therefore excluded as mapping artifacts.

Only three true variants were identified: two corresponded to previously known mutations in the mutants used as positive control strains M1⁻ and M1⁺, validating the sensitivity of our variant calling pipeline. The third was a T>C substitution located ~200 bp upstream of PS15p_212321, which encodes a predicted DNA transposon. This variant was found in E12⁺, an epimutant with confirmed *fkbA*-targeting siRNAs. Given its location in a repetitive, non-coding region, this variant is likely either a mapping artifact or biologically irrelevant. Moreover, E12⁺ was not used in any of the genetic crosses, so this variant could not have contributed to epimutation inheritance in the progeny.

The reviewer also raises the possibility that chromatin modifications at loci other than *fkbA* could mediate a trans-acting silencing mechanism. To investigate this, we performed a genome-wide analysis of H3K9me2 distribution and identified genes with ≥90% coverage by H3K9me2. We then compared H3K9me2-enriched and -depleted genes in epimutant strains relative to a naïve wildtype (Extended Data Fig. 8c-e). We hypothesized that such regions might exhibit altered expression relevant to resistance. However, differential expression analysis using DESeq2 did

not reveal significant expression changes in these loci, suggesting that H3K9me2 redistribution is not directly mediating gene expression changes associated with resistance.

Reviewer #4 (Remarks to the Author):

The paper entitled “RNAi epimutations conferring antifungal drug resistance are inheritable” tackles the challenge to demonstrate the heritable nature of RNAi epimutations in the context of antifungal resistance, which is a question of major importance and an ambitious objective. The model is the human pathogen *Mucor circinelloides*. The main message conveyed by the study is that spontaneous RNAi epimutations in response to drug challenge can be transmitted to the progeny, inheritance being determined only via smRNAs.

The findings are convincing and the message is quite exciting. The set up may appear simple and straightforward but the work performed is definitely a lot. That being said, there are several points that must be addressed in my opinion.

We thank the reviewer for their positive and thoughtful feedback, and for taking the time to carefully evaluate our study. We appreciate the recognition of both the significance and ambition of our work, and we value the constructive comments that assisted us in further improving the manuscript.

In a general manner, the materials and methods section is missing many pieces of information to really allow assessing all technical aspects of the paper, and even less to be re-used if one would wish to. For example, parameters used with the various genome assembly and QC tools are mostly missing, the source and version of *M. lusitanicus* proteome is not mentioned, same with the genetic variation analyses and the ChIP-seq analyses. Also, I’m not sure the protocol used to provoke epimutations is crystal clear. The legend of Fig1a must be extended and/or the text clarified. I may not be the only one being a bit uncertain...

We appreciate the reviewer’s feedback on the Materials and Methods section. We have expanded this section extensively to include key details such as the parameters and commands used for genome assembly and quality control tools, and we have now specified the version and source of the *Mucor lusitanicus* MU402 genome and proteome used in our analyses. We have also clarified the variant calling and ChIP-seq pipelines, including software commands, arguments, and thresholds. In the drug susceptibility testing section, we now provide a clearer description of how resistant colonies were selected and passaged to assess epimutation stability or reversion. Furthermore, we have extended the legend of Fig. 1a to clarify the experimental protocol for inducing epimutations, including steps for drug challenge, colony isolation, and phenotyping. We

have also linked to appropriate Extended Data Fig. 1c and d, which partially explain the protocol. We have also reviewed all figure legends to ensure that relevant methodological or clarifying details are included where appropriate.

Providing details is particularly important in the case of the sRNA sequencing analyses, for which the authors used the Qiagen library system and thus had to apply some specific preprocessing steps to the raw sequencing reads prior use. Similarly, providing the bioawk commands that were used would be welcome for best open science practice (what “specific siRNA features did you use”). This is important since you are using these analyses to make the conclusion that the RNAi machinery works.

We appreciate the reviewer’s emphasis on transparency and reproducibility. In response, we have expanded the Materials and Methods section to include additional details regarding the preprocessing of small RNA sequencing data, including the specific steps required for libraries generated with the Qiagen system. This includes adapter trimming, read filtering by size and quality, and alignment parameters. We also now provide the exact bioawk commands and criteria used to extract siRNA-specific features such as size range (21–24 nt), strand specificity, and genomic location.

Lines 86-87, the authors state that sanger sequencing found genetic mutations in *fkbA* in four isolates out of the 13 tested. However, no information is given regarding what section of the gene was examined, meaning were UTRs checked in addition to cds for example? Non-coding mutations in regulatory elements may still be of influence and the possibility should not be ignored before stating “no DNA sequence changes were found” (lines 87-88).

Thank you for this thoughtful comment. We fully agree with the reviewer that mutations in UTRs and regulatory regions can influence gene function and should not be overlooked. To clarify this point, we have updated the text to specify that both the coding sequence and flanking non-coding regions of *fkbA* were analyzed by Sanger sequencing. To support this, we have added a schematic of the *fkbA* locus showing the amplified regions and primer positions (now included as Extended Data Fig. 1f). Additionally, we performed whole-genome sequencing of resistant isolates and confirmed that no mutations were detected in or near the *fkbA* locus, as shown in the updated variant calling analysis (Extended Data Fig. 7). These data strengthen our conclusion that resistance in these strains is not due to DNA sequence changes in *fkbA* or genome-wide.

Regarding the northern blot anti-*fkbA* antisense, I understand from the legend that it was obtained in the Ex isolates BEFORE their reversion, which makes sense and the sentence lines 90-92 should be more explicit. Did you perform the same northern after reversion? Fig 1d and 1e are pretty convincing but not doing the northern when you have all materials to do so does not make sense. And also, there is a typo in the legend where d and e are mislabeled. Nonetheless, I'm mostly ok with your conclusions (maybe the word resulting line 96 is a bit abusive). That being said, I do not understand the last sentence of the paragraph (lines 98 to 100).

We thank the reviewer for pointing out the typographical errors in the figure legend, which have now been corrected. We have clarified in the text (former lines 90–92) that Northern blotting was performed on active epimutant isolates before their reversion. While we also conducted a Northern blot on one revertant progeny (Extended Data Fig. 5h, isolate #14), this was done as part of the exploratory phase of our research to reduce costs. At that stage, Northern blotting was a cost-effective method to assess siRNA levels. However, once we determined that resistance was linked to siRNAs and RNAi, we switched to small RNA sequencing as our primary method. Small RNA sequencing provides far greater sensitivity, accuracy, and the ability to quantify siRNAs, allowing us to map the siRNA distribution across the entire *fkbA* sequence. Although we replicated the Northern blot results with small RNA sequencing to strengthen our conclusions, performing the reverse, i.e., replicating sRNA-seq data with Northern blots was deemed unnecessary for the following reasons: 1) RNA samples were limited due to simultaneous small RNA and mRNA sequencing, and Northern blotting typically requires a large quantity of small RNAs (an entire preparation in most cases); 2) repeating Northern blots would not yield new relevant insights, as our sequencing results already provide comprehensive, robust, and reproducible evidence supporting our conclusions; 3) our existing Northern blot protocols rely on radioactive labeling, and we are making a conscious effort to reduce the use of radioactivity in the lab for reasons of safety (reducing exposure risks and contamination hazards for personnel), sustainability (decreasing hazardous waste generation and the need for specialized disposal procedures), and alignment with best practices in modern molecular biology. While we are aware of non-radioactive labeling alternatives, implementing and validating them at this stage would not be an efficient use of resources. Instead, transitioning to sequencing technologies was deemed the most effective and forward-looking approach, providing both higher resolution data and alignment with modern methodological standards.

In response to the reviewer's comment, we have also revised former line 96 to avoid overstatement, clarifying that increased siRNA abundance aligns with post-transcriptional gene silencing of *fkbA*. Additionally, we have refined former lines 98–100 to better explain our statement.

In the section dealing with chromatin involvement, the authors state too much from their observations. The title line 149 uses the words “exclusively” and “uncoupled from chromatin modifications”. First, there is more to this world than H3K9. Second, although the demonstration is conclusive on the elements they stated (i.e., H3K9me deposition, genetic mutation in *fkbA*, and chromosomal rearrangement) the possibility of other contributing mechanisms cannot be excluded, starting with possible genetic mutations elsewhere than *fkbA*. The “exclusive” wording must be corrected as well as the assimilation of H3K9me to “chromatin modifications”. Same with lines 177-178

We appreciate the reviewer's concern that additional mechanisms, including other chromatin modifications, may contribute to epimutation. H3K9 methylation is the best-characterized transcription-repressive chromatin modification across eukaryotes and was therefore our primary focus. We fully acknowledge, however, that other chromatin marks, known and yet to be discovered, can modulate gene expression.

Mucor circinelloides is quite unusual in this regard. Like most Mucoralean fungi, it lacks the DNA methyltransferases required for 5mC and for H3K27me, two other major chromatin modifications involved in transcriptional gene silencing. In prior work (Navarro-Mendoza et al., 2023, PNAS), we demonstrated that H3K9me and heterochromatin maintenance are independent of the RNAi machinery in this species. Given the marked reduction in *fkbA* mRNA expression, and the absence of other canonical repressive chromatin marks in *Mucor*, we specifically tested H3K9me in the epimutants. We have clarified this rationale in the revised text.

Additionally, we showed that RNA polymerase II (RNAP) occupancy remains unchanged, and *fkbA* is being actively transcribed despite its reduced expression at the mRNA level during epimutation (previously shown in Fig. 3b), which is consistent with posttranscriptional gene silencing. In response to the reviewer's concerns, we have expanded our analysis to appropriately quantify RNAP occupancy differences between wildtype and epimutant strains using DiffBind (Extended Data Fig. 8a, Supplementary Data 3). We have also included new genomic plots showing *fkbA* transcription during epimutation over a more appropriate range (Extended Data Fig. 8b). Overall, these results support posttranscriptional gene silencing rather than transcriptional

gene silencing, providing evidence for similar "open" chromatin states in wildtype and epimutant strains.

Furthermore, we conducted additional variant calling analyses with our WGS data and did not identify off-target mutations in the epimutants that could explain the observed drug resistance (Extended Data Fig. 7).

Lastly, since we cannot test every known chromatin modification, we have revised the wording to be more cautious. We have removed the term "exclusive" and replaced "chromatin modifications" with "heterochromatin formation" in line 149. In lines 177-178, we specify which chromatin marks do not repress *fkfA* transcription.

The discussion is somehow not really discussing the results. Bringing up more critical views about other chromatin marks or the situation in other fungi (machinery-wise for example) would be a nice addition. The last paragraph starting line 242 is plain useless, as a repetition of info already provided in introduction.

We thank the reviewer for their constructive feedback. In response, we have revised the Discussion section to include a more critical comparison of epimutations and their associated chromatin machinery, focusing on *Schizosaccharomyces pombe*, the other fungal system where epimutations have been clearly characterized, and compare the distinct mechanisms underlying RNAi-mediated gene silencing in these organisms.

We also agree that the original closing paragraph was partially redundant with the Introduction. We have therefore modified it to a more concise closing statement that underscores the implications of our findings for understanding epigenetic regulation, fungal adaptability, and antifungal resistance, without overtly repeating prior content.

Regarding data availability, the reviewer link failed to work in my hands. But that may be a server glitch. In any case, the "will be" must be replaced by "is" in the text, assuming the author is talking to a reader.

We have replaced "will be" with "is" in the manuscript, anticipating that the data will be available just before publication. Additionally, we have regenerated the reviewer access tokens to ensure data is accessible to reviewers beforehand. The following updated links have been verified as active and functional by independent laboratory members:

- PRJNA1168935: PS15- genome assembly and raw ONT and Illumina reads at <https://dataview.ncbi.nlm.nih.gov/object/PRJNA1168935?reviewer=7tarapo67fv488971s3f45h9k1>
- PRJNA1168941: PS15+ genome assembly and raw ONT and Illumina reads at <https://dataview.ncbi.nlm.nih.gov/object/PRJNA1168941?reviewer=rc28mtdkmqrf28ls6klr4ckgkj>
- PRJNA1170303: remaining raw reads, including small RNA, mRNA, ChIP, and WGS data at <https://dataview.ncbi.nlm.nih.gov/object/PRJNA1170303?reviewer=ir6sn54rqvqdc3cua35r9q71ee>

I also have minor comments.

– Extended Figure 1d is too small to see the colonies marked by red arrows;

We thank the reviewer for pointing this out. In Extended Figure 1d, we have modified the layout to include a zoomed-in view of the relevant colonies. This adjustment makes the colonies marked by red arrows more easily visible, ensuring that key details can be seen clearly without the need for magnification.

– No picture of M2+ and M3+ are provided

Extended Data Fig. 1e now shows M2+ and M3+ growth in FK506 and rapamycin.

– Lines 80-81: missing references for the targets of rapamycin and FK506

A single study identified both targets in *Mucor* spp. and is now referenced accordingly.

– Line 115 fig a c is d e

We have revised line 115 to clearly indicate the corresponding figures. For FK506 plating, we refer to Extended Figure 2a–c, and for FK506 and rapamycin plating, we refer to Figure 2a–c.

– Line 173 fig 3e is 3d

We have revised line 173 and we now refer to Figure 3d.

– Line 125: add p value

Lines 124-126 contain a brief summary of overall results and specific *p*-values were provided above that sentence.